# Host-derived organic acids enable gut colonization of the honey bee symbiont *Snodgrassella alvi*

Andrew Quinn ⬚ [1,5], Yassine El Chazli ⬚ [1,5], Stéphane Escrig[2], Jean Daraspe[3], Nicolas Neuschwander[1], Aoife McNally[1], Christel Genoud[3], Anders Meibom ⬚ [2,4] & Philipp Engel ⬚ [1]✉

Diverse bacteria can colonize the animal gut using dietary nutrients or by engaging in microbial crossfeeding interactions. Less is known about the role of host-derived nutrients in enabling gut bacterial colonization. Here we examined metabolic interactions within the evolutionary ancient symbiosis between the honey bee (*Apis mellifera*) and the core gut microbiota member *Snodgrassella alvi*. This betaproteobacterium is incapable of metabolizing saccharides, yet colonizes the honey bee gut in the presence of a sugar-only diet. Using comparative metabolomics, [13]C-tracers and nanoscale secondary ion mass spectrometry (NanoSIMS), we show in vivo that *S. alvi* grows on host-derived organic acids, including citrate, glycerate and 3-hydroxy-3-methylglutarate, which are actively secreted by the host into the gut lumen. *S. alvi* also modulates tryptophan metabolism in the gut by converting kynurenine to anthranilate. These results suggest that *S. alvi* is adapted to a specific metabolic niche in the honey bee gut that depends on host-derived nutritional resources.

Gut bacteria and their hosts typically engage in mutualistic interactions. Metabolic exchange from the gut microbiota to the host is vital for uptake of essential nutrients, gut health and immune system function[1]. In turn, bacteria profit from a stable niche environment and frequent supply of exogenous food. The role of host-secreted metabolites that benefit bacteria in the gut is less well understood. Such metabolic exchange is difficult to identify due to the overwhelming contributions of the diet and microbial products towards gut metabolites.

Deciphering the extent to which host metabolite secretion drives microbial colonization of native symbionts is aided by a simple, tractable system in which diet and microbiota-derived metabolites can be tightly controlled. The Western honey bee (*Apis mellifera*) provides such a system. Its gut microbiota is broadly stable and composed of only eight to ten genera[2,3] many of which have longstanding evolutionary

associations with their host that date back to the emergence of social bees >80 million years ago[4–6]. Bacteria of these genera are culturable and can be inoculated individually or as defined communities into gnotobiotic bees[7]. Furthermore, while the honey bee diet features a rich mixture of compounds found in nectar and pollen, bees can survive for extended periods on pure sugar water diets[8].

Most members of the bee gut microbiome are primary fermenters, possessing a broad range of carbohydrate degradation enzymes that enable them to use hemicellulose, pectin, starch or glycosides found in pollen[3,9]. A notable exception is the Betaproteobacteria *Snodgrassella alvi*. It colonizes the cuticular surfaces of the ileum in the hindgut and displays a markedly different metabolism[10,11]. Lacking a functional glycolysis pathway, *S. alvi* profits from acids in the gut, generating energy from an aerobic tricarboxylic acid (TCA) cycle and biomass through

[1]Department of Fundamental Microbiology, University of Lausanne, Lausanne, Switzerland. [2]Laboratory for Biological Geochemistry, Ecole Polytechnique Fédérale de Lausanne (EPFL), Lausanne, Switzerland. [3]Electron Microscopy Facility, University of Lausanne, Lausanne, Switzerland. [4]Center for Advanced Surface Analysis, Institute of Earth Sciences, University of Lausanne, Lausanne, Switzerland. [5]These authors contributed equally: Andrew Quinn, Yassine El Chazli. ✉e-mail: philipp.engel@unil.ch

gluconeogenesis. Fermentation by other microbial members has been proposed as the primary source of short chain fatty acids (SCFAs) consumed by *S. alvi*. In particular, bacteria of the gammaproteobacterial genus *Gilliamella* are probably mutualistic partners for metabolic crosstalk, as they colocalize with *S. alvi* within biofilms attached to the cuticular surface in the ileum and share complementary metabolic capabilities[10]. Experimental evidence from in vitro growth experiments bolstered this hypothesis, showing that *S. alvi* grows on the spent media of *Gilliamella*, while consuming numerous products of the *Gilliamella* metabolism, such as succinate and pyruvate[12].

Although a strong case can be made for niche exploitation through bacterial crossfeeding, this does not explain previous results in which *S. alvi* was able to monocolonize bees fed diets of sugar water and pollen[12]. To better understand the nutrient sources that *S. alvi* exploits, we provided bees with a simple (sugar water) or complex (sugar water and pollen) diet and colonized them with *S. alvi* alone or together with divergent strains of the genus *Gilliamella*. Surprisingly, we found that a simple sugar water diet was sufficient for *S. alvi* to colonize the honey bee gut. Subsequent metabolomics analysis indicated that host-derived carboxylic acids enable *S. alvi* colonization. We validated this with a series of experiments showing that (1) these carboxylic acids are synthesized by the host, (2) *S. alvi* uses them for growth and (3) the findings hold across a range of divergent *Snodgrassella* strains and species.

## Results

### *S. alvi* colonizes the gut with only sugar in the diet

We colonized microbiota-free (MF) honey bees with *S. alvi* (strain wkB2) or a mixture of *Gilliamella* strains (strains wkB1, ESL0169, ESL0182, ESL0297) or with both phylotypes together and provided a diet composed of only sterile-filtered sugar water (sucrose) or sugar water and sterilized polyfloral pollen for 5 days (Fig. 1a). Colonization levels of both phylotypes were assessed by quantitative polymerase chain reaction (qPCR) and colony-forming unit (c.f.u.) plating 5 days postinoculation (Fig. 1b and Extended Data Fig. 1). Surprisingly, *S. alvi* colonized equivalently in the guts across all treatment groups, that is, sugar water in the diet was sufficient to enable *S. alvi* colonization and neither the addition of pollen to the diet nor cocolonization with *Gilliamella* significantly increased *S. alvi* loads (two-sided Wilcoxon rank sum test with *S. alvi* (sugar water) as reference group). Importantly, qPCR analysis with universal bacterial and fungal primers showed no or very low levels of amplification in most samples, apart from gut samples containing pollen which resulted in relatively high background amplification with the universal bacterial primers as previously reported (Extended Data Fig. 1c,d)[12,13]. Culturing of gut homogenates of MF bees on different media resulted in no microbial growth in any of the tested conditions (Methods). Hence, we can rule out that systematic contaminations with other microbes facilitated *S. alvi* colonization through crossfeeding in the monocolonization treatment.

### *S. alvi* depletes organic acids in the honey bee gut

To search for putative *S. alvi* growth substrates originating from the host, the pollen diet or *Gilliamella*, we next extracted metabolites from the midgut and hindgut for a subset of the colonized bees and analysed them via gas chromatography–mass spectrometry (GC–MS). The presence of pollen in the gut significantly increased the abundance of nearly half of the annotated metabolites (125 of 233) in MF bees (Extended Data Fig. 2). Thus, we compared results between colonized and MF bees only within each dietary treatment. When we examined monocolonization of *S. alvi* in bees fed with sugar water, we identified many carboxylic acids, including citrate, 3-hydroxy-3-methylglutarate (3Hmg) and glycerate that were significantly less abundant than in the MF controls (Fig. 1c). These carboxylic acids, along with others (for example, 2-ketoisocaproate, *α*-ketoglutarate and malate) were also depleted in *S. alvi* monocolonized bees that were fed pollen (Fig. 1c), despite the

substantial differences in the overall metabolite profiles between the two dietary conditions. In contrast, few metabolites were significantly more abundant in *S. alvi* colonized bees relative to MF bees, with only anthranilate, a product of tryptophan metabolism, accumulating in colonized bees of both dietary treatments (Fig. 1c).

We then fit each metabolite with a linear mixed model to quantify how strongly metabolite changes were influenced (that is, the fixed effect) by each of the three independent experimental variables (that is, the presence of *Gilliamella, S. alvi* or pollen) across the eight experimental conditions, This crossconditional analysis confirmed that colonization with *S. alvi* resulted in significantly (*P* < 0.05; see link given in the 'Code availability' for supporting data) decreased abundances of carboxylic acids, particularly 3Hmg, citrate, malate, fumarate and glycerate (Fig. 1d). We could also infer which metabolites were co-produced, co-consumed or cross-fed (and in which direction) between the two bacteria (Fig. 1d). For example, we found evidence supporting our initial hypothesis of metabolic cross-feeding from *Gilliamella* to *S. alvi* in the form of lactate, pyruvate and other unknown compounds that were more abundant with *Gilliamella* and less abundant with *S. alvi* (Fig. 1d and Extended Data Fig. 3). However, we found no evidence of reverse cross-feeding from *S. alvi* to *Gilliamella*. Instead, we found that both species compete for certain metabolites and act synergistically to synthesize others. Competition or co-consumption, centred on the carboxylic acids, as colonization with *Gilliamella* also led to depletion of citrate and glycerate and to a lesser extent, malate and fumarate (Fig. 1d and Extended Data Fig. 3). Cooperative synthesis revolved around four metabolites: acetate, succinate, benzoate and unknown_12.74. Of these, only acetate was also more abundant in monocolonization (*Gilliamella*) than in MF controls (Fig. 1d and Extended Data Fig. 3). Finally, nearly all the metabolites depleted with *S. alvi* colonization were positively affected by pollen, except for citrate, glycerate, acetate and two unknown compounds, which were unaffected, and urea, whose abundance was negatively correlated with pollen. Taken together, these results show that each of the tested variables affects nutrient availability and metabolism of *S. alvi*, although metabolic crossfeeding and a nutrient-rich diet do not lead to increased colonization levels in the gut.

### Host sugar catabolism provides *S. alvi* substrates in vivo

We next sought to measure host production of carboxylic acids in the guts of MF bees over the first 6 days postemergence and compare the abundances to what the bee could have extracted from the average amount of pollen consumed (24.3 ± 7.2 mg per bee) as estimated on the basis of data collected from the previous experiment (Methods, Extended Data Fig. 4 and Supplementary Table 1). Moreover, to rule out microbial or fungal contamination as a source of these compounds, we rigorously checked once more for live contamination by plating the guts of newly emerged (day 0) and 6-day-old MF bees on eight different media in three different growth environments (Methods), finding no evidence of live bacterial or fungal contamination (*n* = 8). Even though the bees only consumed sugar water, we found that many gut metabolites, including glycosylamines, sugar alcohols and carboxylic acids, were significantly more abundant 6 days after emergence (Extended Data Fig. 5). Focusing specifically on the compounds depleted in the presence of *S. alvi*, we found that citrate was most abundant in the gut, with its concentration increasing from 29 ± 14 µmol mg$^{-1}$ at day 0 to 73 ± 34 µmol mg$^{-1}$ at day 6 post-emergence. Substantially less citrate, 3 ± 2 µmol mg$^{-1}$, was extracted from pollen (Fig. 2a). A similar trend was also found for 3Hmg, *α*-ketoglutarate, isocitrate and kynurenine. Of *S. alvi's* putative substrates, only caffeate and lactate were more abundant in pollen than in the bee gut. The increasing abundance of many compounds over 6 days, as well as their generally low abundance in pollen suggests that active metabolism by the host substantially impacts the gut metabolome. Specifically, easily digestible nutrients in the diet are absorbed and metabolized by the bee upstream of the

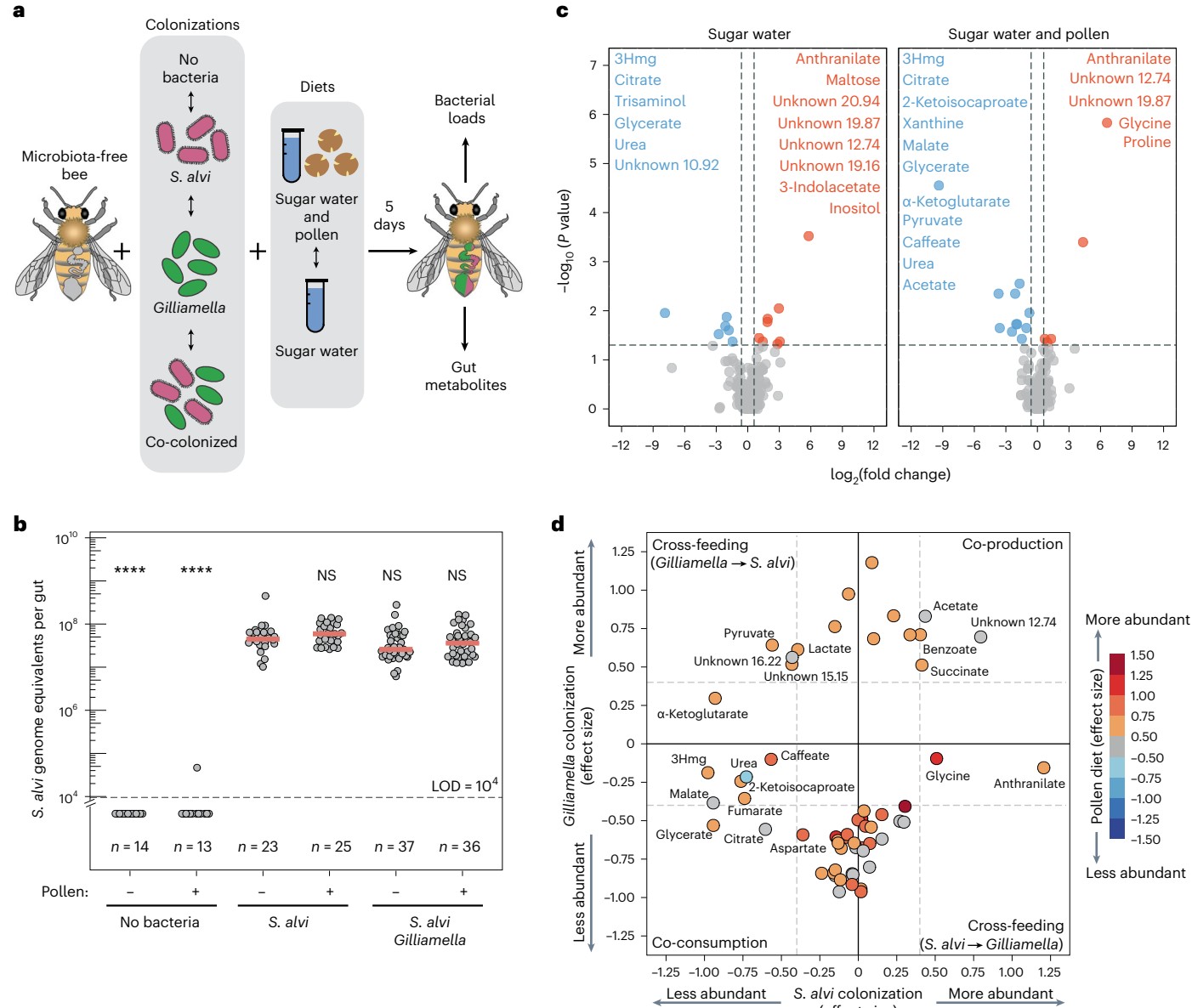

**Fig. 1 | _S. alvi_ colonizes bees fed with sugar water only and depletes host, pollen and _Gilliamella_-derived metabolites. a**, Outline of the bee colonization experiment with four colonization treatments (no bacteria, _S. alvi_, _Gilliamella_ or co-colonized) and two dietary treatments (sterile sugar water and pollen or sterile sugar water only), repeated with bees from five hives. Five days after colonization gnotobiotic bees were sacrificed and the bacterial load and the metabolites in the gut quantified. **b**, Number of _S. alvi_ cells (number of bacterial genome equivalents) per bee gut determined by qPCR using _S. alvi_-specific 16S rRNA gene primers. Each value represents the number of cells in a single bee gut, red bars represent the median, horizontal black line represents qPCR limit of detection (LOD) $< 10^4$. For each treatment, the presence/absence of pollen in the diet is indicated by ± sign. Significance determined by a two-sided Wilcoxon rank sum test with _S. alvi_ (pollen −) as reference group (no bacteria, ****$P = 2.37 \times 10^{-7}$; no bacteria (pollen), ****$P = 5.73 \times 10^{-7}$). **c**, Significantly differentially abundant gut metabolites between _S. alvi_ colonized (pollen ±, $n = 11/12$) and MF bees (that is, 'no bacteria' treatment) (pollen ±, $n = 12/10$). Significantly depleted and produced metabolites are shown in blue and orange, respectively. Adjusted

$P$ values were calculated using a two-sided Wilcoxon rank sum test, with Benjamini–Hochberg (BH) correction. **d**, Results from linear mixed-effects modelling show the contributions (fixed effect sizes) of microbial colonization and diet towards $z$-score normalized metabolite abundances. Only metabolites significantly correlated with _S. alvi_ colonization are named ($n = 95$, linear mixed-effects models fitted by REML with bee colony as random effect, $P$ values adjusted for multiple testing with BH correction). Large values (±) indicate a strong correlation between a factor and a corresponding change in metabolite abundance. Values centred around zero (inside the dotted lines) are not significant. The effect size of pollen is indicated by the colour of each metabolite, red indicates significantly positive correlation, while blue is a significant negative correlation. The four corners of the graph represent the four types of metabolic interactions between _S. alvi_ and _Gilliamella_ (from the top left: cross-feeding from _S. alvi_ to _Gilliamella_, co-production by both bacteria, cross-feeding from _Gilliamella_ to _S. alvi_, co-consumption). Carboxylic acids are grouped principally on the lower-left section of the plot indicating that they are less abundant with _S. alvi_ and _Gilliamella_ colonization. NS, not significant.

hindgut, while downstream metabolic products are excreted into the hindgut and serve as substrates for _S. alvi_.

To corroborate this hypothesis, we fed MF bees for 6 days with 45% of the dietary sucrose replaced by 100% [U-$^{13}$C$_6$] glucose. Half of

the bees were also provided standard pollen to assess its contribution relative to simple sugars in the synthesis of gut metabolites. We then analysed the resulting $^{13}$C-isotopic enrichments in gut metabolites with a focus on those indicated as substrates for _S. alvi_. The carboxylic

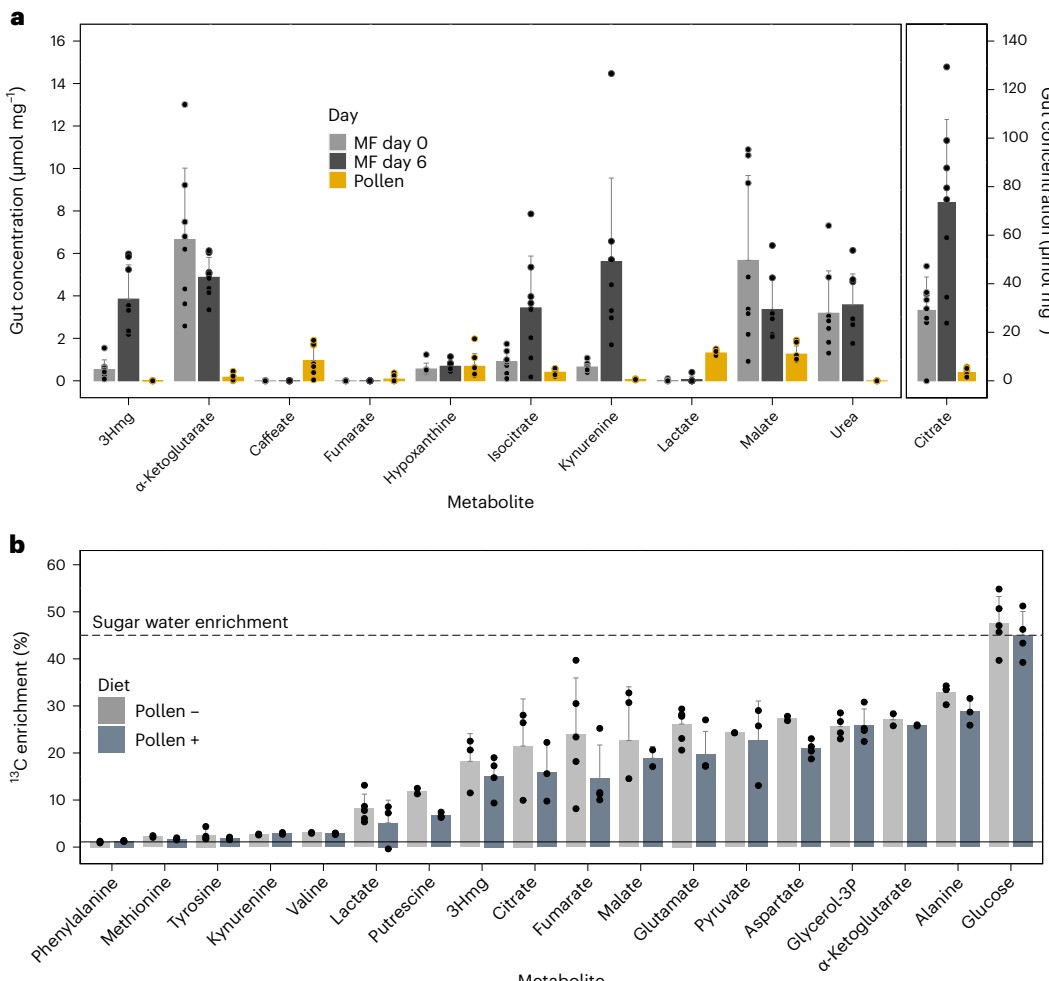

**Fig. 2 | Gut metabolite abundances and isotope labelling reveal the host origin of _S. alvi_ substrates. a**, Abundances of selected metabolites in the guts of MF bees increase from emergence (day 0) to adult age (day 6). All metabolites except for caffeate and lactate were less abundant when extracted directly from an equivalent amount of pollen consumed per bee over 6 days. All bees were from the same hive. Quantitates from pollen extract were adjusted on the basis of average total pollen consumption per bee (Methods). Bars represent mean + s.d. for each metabolite, $n = 8$ for each condition. **b**, The $^{13}$C metabolite enrichment (%) in bee guts shows that carboxylic acids and non-essential amino acids

become highly labelled on feeding [U-$^{13}$C$_6$] glucose to MF bees. Dark and light bars indicate, respectively, presence and absence of pollen in the diet. The solid line denotes the natural $^{13}$C enrichment (1.108%), while the dashed line denotes the $^{13}$C enrichment of sugar water solution fed to bees. Bars show the mean + s.d. from $n = 4$ bee guts that were analysed for both dietary groups. All bees were from the same hive. A single mass feature was manually selected and plotted for metabolites with many analysed features, with a preference for features containing the most carbon atoms. See 'Code availability' section for data quality filtering steps.

acids and non-essential amino acids were significantly $^{13}$C-enriched in the bee gut. In contrast, other compounds, such as kynurenine and essential amino acids did not show any enrichment (Fig. 2b). The lack of isotope labelling indicates that these compounds were not actively synthesized from glucose by adult bees during the first 6 days postemergence. Instead, they were either leftover in the gut from the larval development stage or were acquired from the larval diet, such as in the case of essential amino acids and their catabolic products. As expected, the average $^{13}$C enrichment levels of most metabolites dropped in bees fed with pollen compared to those fed with sugar water only. The $^{13}$C dilution can occur directly from non-labelled metabolites in pollen, as well as indirectly from host metabolism of pollen substrates. However, the average $^{13}$C enrichment of carboxylic acids only dropped from $21 \pm 6\%$ to $17 \pm 7\%$, which serves as further evidence that host metabolism of simple sugars rather than dietary consumption is the predominant source of carboxylic acids in the gut (Fig. 2b). We thus conclude from this analysis that the carboxylic acids used by _S. alvi_ are mostly de novo synthesized from sugar metabolism of the host.

**NanoSIMS reveals transfer of host compounds to _S. alvi_**

While our previous experiments suggested host nutrient foraging by _S. alvi_, the results did not provide direct evidence that _S. alvi_ assimilates biomass from these compounds. Therefore, we probed the flow of metabolites from the bee to _S. alvi_ using a 'pulse–chase' $^{13}$C-isotope labelling experiment and then measured the enrichment within _S. alvi_ cells and surrounding host tissue using nanoscale secondary ion mass spectrometry (NanoSIMS) complemented with measurements of metabolite enrichments using GC–MS (Fig. 3 and Supplementary Table 3). To do so, we enriched newly emerged MF bees in $^{13}$C by feeding them 100% [U-$^{13}$C$_6$] glucose for 4 days after emergence. We then inoculated them with _S. alvi_ and waited 1 day before switching their diet to naturally abundant, 98.9% $^{12}$C glucose (Fig. 3a). This ensured that the largest _S. alvi_ population increase, between 24- and 48-h post-colonization, occurred without a dietary $^{13}$C source but still in a highly $^{13}$C-labelled environment (Fig. 3b).

Accordingly, we measured rapid $^{13}$C-labelling turnover in _S. alvi_ in 18 images coming from two bees (sampled at 48 and 72 h postinoculation) in which we could detect bacterial cells (Fig. 3c,e and Source

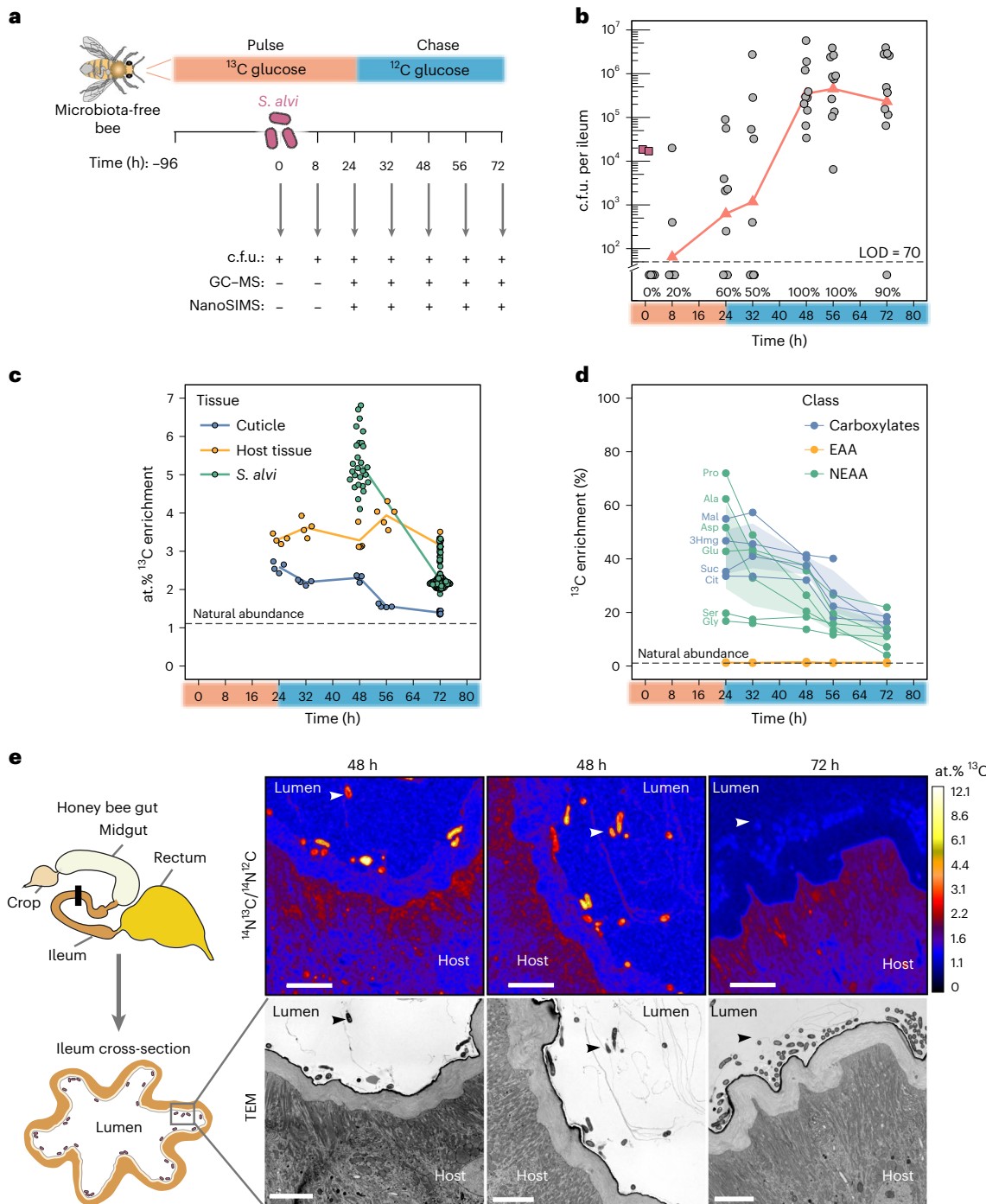

**Fig. 3 | S. alvi builds biomass from substrates derived from the glucose metabolism of the host. a**, Experimental design of the pulse–chase isotope labelling gnotobiotic bee experiment. Newly emerged MF bees were fed 100% ¹³C glucose for 4 days (96 h), colonized with a defined quantity of *S. alvi* and then the ¹³C glucose was replaced by ¹²C glucose 24 h after colonization. The timeline indicates when bees were sampled for c.f.u. plating, GC–MS and NanoSIMS analysis relative to the timepoint of colonization (0 h). All bees were from the same hive. **b**, Colonization levels of *S. alvi* in the ileum until 72 h postinoculation show the initial strong colonization bottleneck and subsequent exponential increase in cell numbers. Colonization success is indicated as percentage of bees with detectable c.f.u. Pink squares indicate the inoculum of *S. alvi* (OD = 0.1). The LOD of 70 c.f.u. per ileum is indicated with the dashed line. **c**, The average at.% ¹³C enrichments of *S. alvi* cells decreases rapidly, while the enrichment of host cells and epithelium remains constant over 72 h in ileum cross-sections imaged with NanoSIMS. Each data point represents the mean of ROI; bacterial ROIs consist of single bacterial cells, while epithelium and host cell ROIs represent the total area

encompassing epithelium or host cell tissue in each image. ROI raw counts and calculated enrichments are listed in Supplementary Table 3. The dashed line at 1.1% indicates the natural ¹³C abundance calculated from NanoSIMS images of ¹²C control bees. **d**, ¹³C enrichment of metabolites in the gut steadily decreases in the ¹²C chase phase. Dots represent the average ¹³C enrichment of a given metabolite across the bees sampled at that timepoint. Average values and colours are shown by metabolite class: carboxylic acids (blue), non-essential amino acids (NEAA, green), essential amino acids (EAA, yellow). Shaded regions encompass the maximum and minimum limits of each metabolite class, *n* = 3 bees per timepoint. Measured metabolites are listed in Supplementary Table 3. The dotted line at 1.1% indicates the natural ¹³C abundance. **e**, Schematic drawing of the honey bee gut shows the region in the ileum (black bar) where cross-sections were taken, partially adapted from ref. 12. TEM images and corresponding NanoSIMS images of two different ¹³C-labelled bees at 48 and 72 h after inoculation. White arrows indicate *S. alvi* cells. The at.% ¹³C represents percentage of ¹³C atoms; the natural ¹³C abundance is ~1.1 at.%. Scale bars, 5 μm.

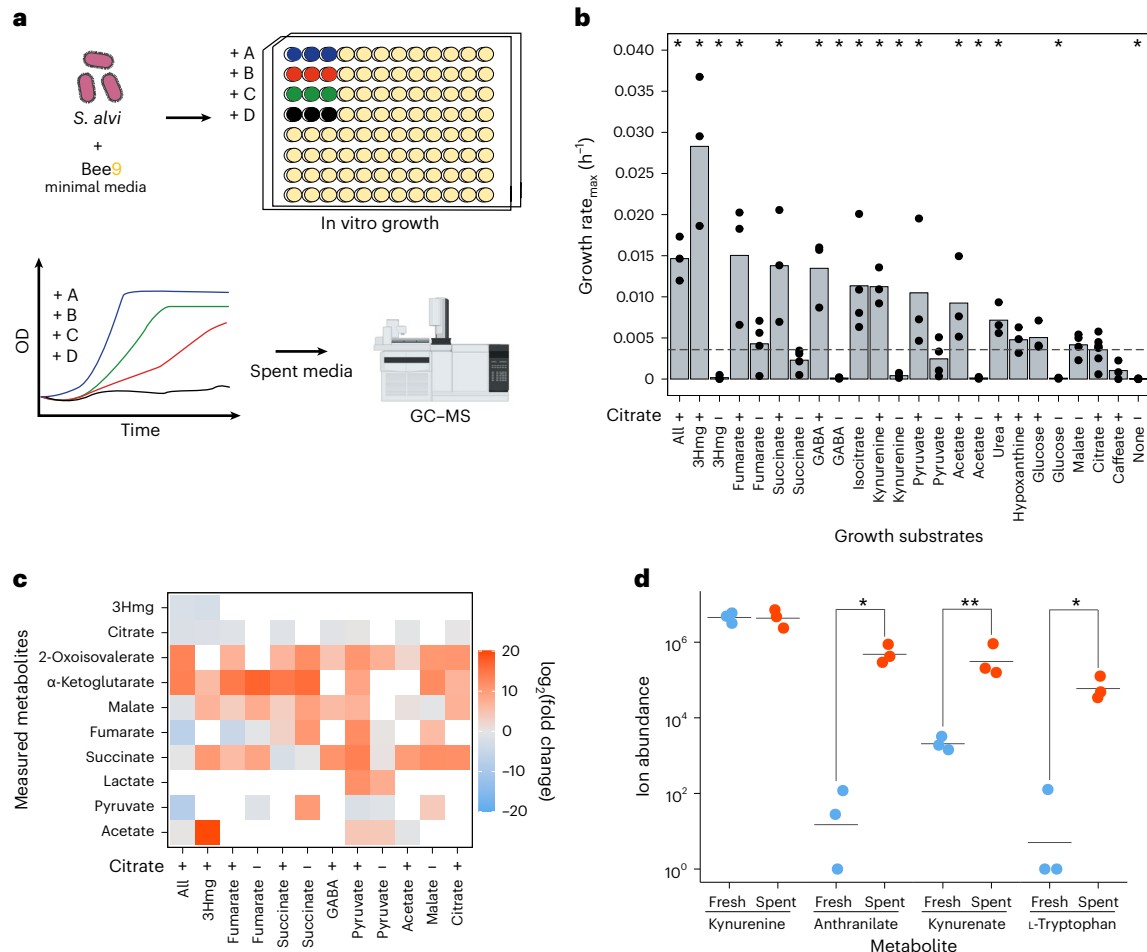

**Fig. 4 | Host-synthesized compounds enhance *S. alvi* growth in vitro.**
**a**, Schematic diagram depicting growth and spent supernatant analysis. The Bee9 medium was derived from M9 base medium (Supplementary Information). **b**, The maximum growth rate across single carbon sources with (+) or without (−) citrate added to the medium, reveals that growth on citrate + 3Hmg was faster than for all other conditions, including 'All', where all substrates were pooled together. The average value (bars) was calculated from independent biological replicates (citrate (+), $n = 3$; citrate (−), $n = 4$). Adjusted significance values were calculated using two-sided Wilcoxon rank sum test with BH correction, using the growth on citrate (+) as the reference state (dotted line), *$P < 0.05$

(Supplementary Table 3). **c**, Abundances of carboxylic acids in the spent supernatant, relative to growth on citrate alone, vary with growth substrates. While some carboxylic acids including α-ketoglutarate and malate are produced across most conditions, acetate is a growth substrate but only produced with either pyruvate or 3Hmg in the media. Average fold changes calculated from $n = 3$ replicates. **d**, Addition of kynurenine to the medium leads to the production of anthranilate, kynurenic acid and L-tryptophan. Significant changes between fresh and spent media were calculated using a two-sided, paired *t*-test from $n = 3$ independent samples (anthranilate, *$P = 0.015$; kynurenate, **$P = 0.004$; L-tryptophan, *$P = 0.039$). GC–MS schematic in **a** created with BioRender.com.

Data of Fig. 1). The $^{13}$C enrichment was substantially higher in *S. alvi* cells than in the adjacent host ileum tissue 48 h after inoculation but then dropped below the levels of host tissue 72 h after inoculation (at.% = start, 5.56; end, 2.25) (Fig. 3c). In contrast, the ileum tissue showed a constant $^{13}$C enrichment over the course of the experiment (at.% = start, 3.31; end, 3.16), whereas the cuticle lining of the gut epithelium was comparatively less enriched initially and continued dropping to almost natural $^{13}$C enrichment levels after 72 h (at.% = start, 2.58; end, 1.39) (Fig. 3c).

The $^{13}$C enrichment of carboxylic acids and host-synthesized (non-essential) amino acids in the gut dynamically shifted in response to the $^{13}$C/$^{12}$C switch (Fig. 3d and Supplementary Table 3). Initially, they were highly $^{13}$C-enriched ($38 \pm 14.7\%$) but the labelling fell steadily to approximately half the original level ($14 \pm 5.8\%$) within 48 h (Fig. 3d). As a control, we found that essential amino acids (non-host-synthesized) were not enriched above the level of natural $^{13}$C abundance (1.11%) throughout the experiment. Thus, these results show that products of host glucose metabolism are actively used by *S. alvi* to build its biomass during early colonization.

## In vitro growth assays confirm active metabolism by *S. alvi*

We next tested whether the putative host-synthesized substrates were sufficient for growth of *S. alvi* as sole carbon sources in a chemically defined liquid medium 'Bee9', which we derived from standard M9 media (Fig. 4a and Supplementary Information). We used Bee9, after finding that *S. alvi* consumes amino acids present in standard M9 previously used to grow this bacterium[12]. Only three metabolites (citrate, isocitrate and malate) supported growth as sole carbon sources in this condition. However, the addition of 3Hmg, fumarate, succinate, gamma-aminobutyrate (GABA), kynurenine or urea to Bee9 + citrate significantly improved the maximal growth rate of *S. alvi* relative to Bee9 + citrate alone (Fig. 4b). In particular, 3Hmg and GABA had dramatic effects on growth rates, increasing them $3.52 \pm 0.10$ and $2.76 \pm 0.04$-fold, respectively.

Analysis of the spent media revealed that all tested metabolites, except for the glucose negative control, were depleted by *S. alvi* (Extended Data Fig. 6). This further confirmed that the identified host-derived organic acids are growth substrates of *S. alvi*. Unlike the results from our colonization experiments, *S. alvi* produced numerous

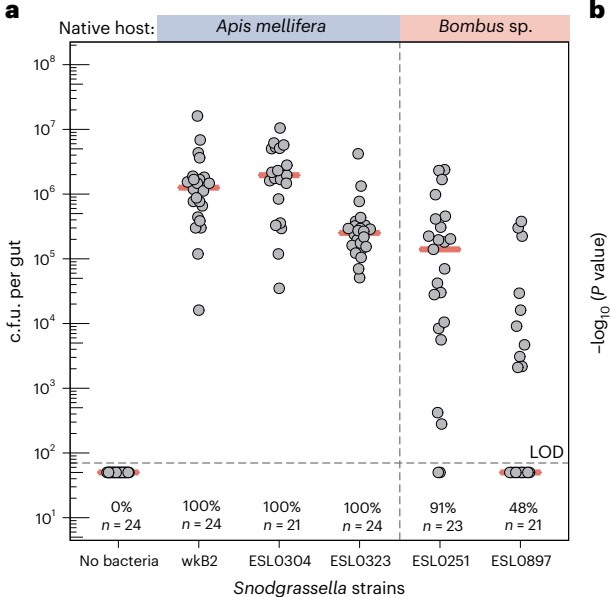

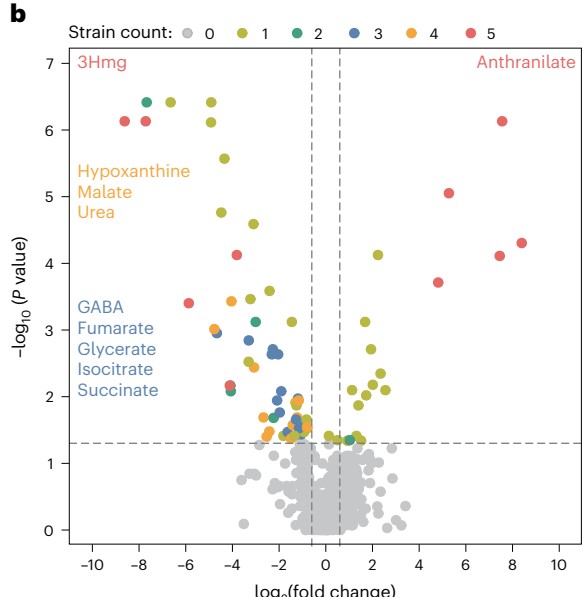

**Fig. 5 | Gut colonization and depletion of host-derived metabolites is conserved across divergent *Snodgrassella* strains. a**, Colonization levels of divergent *Snodgrassella* strains in monocolonized bees fed sugar water. Pink bars denote the median of c.f.u. per gut for each group. The LOD < 70 colonies per gut is shown with a horizontal dashed line. The top bar indicates *Snodgrassella* strains native to *A. mellifera* versus *Bombus* sp. Colonization success (%) is shown for each strain, defined as guts with detectable c.f.u. counts. The experiment was repeated with bees from two hives. **b**, Volcano plot of gut metabolites showing the similarities in metabolite changes from MF versus colonized bees across strains. Each metabolite is plotted five times (once per *Snodgrassella* strain) and colour coded by the number of strains in which it is significantly differentially abundant versus MF bees. Only 3Hmg and anthranilate were significantly depleted or enriched by all five strains, while four of the strains depleted hypoxanthine, malate and urea and three of the five depleted GABA, fumarate, glycerate, isocitrate and succinate. Adjusted significance values were calculated using a two-sided Wilcoxon rank sum test, with BH correction (MF, *n* = 12; wkB2, *n* = 12; ESL0304, *n* = 12; ESL0323, *n* = 13; ESL0251, *n* = 17; ESL0897, *n* = 12).

carboxylic acids, which varied on the basis of the carbon source. For example, acetate was a metabolic substrate but it was also produced in the presence of pyruvate or 3Hmg. Meanwhile, $\alpha$-ketoglutarate, 2-oxoisovalerate, malate and succinate were produced across most growth conditions (Fig. 4c). Consistent with the in vivo metabolomics, we detected production of anthranilate when kynurenine was added to the media, and we further measured the production of kynurenic acid and tryptophan (Fig. 4d). *S. alvi* wkB2 encodes a putative kynureninase gene (SALWKB2_0716, Genbank genome: CP007446; ref. [10]) that is probably responsible for the conversion of kynurenine into anthranilate. Interestingly, this gene seems to have been acquired by horizontal gene transfer as indicated by a phylogenetic analysis (Extended Data Fig. 7): apart from other *Snodgrassella* strains, the closest sequences of the kynureninase gene were not found in other Neisseriaceae but in more distantly related Betaproteobacteria (genera *Pusilimonas* and *Alcaligenes*) and in Gammaproteobacteria (genera *Ignatzschineria* and *Acinetobacter*).

### Colonization is conserved across the *Snodgrassella* genus

As a final step, we examined whether our results were unique to the *S. alvi* wkB2 type strain or were more generally applicable to the genus, by inoculating bees with five divergent strains of *Snodgrassella*, three native to honey bees and two native to bumble bees. We also lengthened the colonization experiment from 5 to 10 days to account for a potential delay in colonization of the honey bee gut by non-native strains. All five strains successfully colonized bees fed only sugar water. However, the efficiency and extent of colonization varied between strains. Strains of *Snodgrassella* native to honey bees colonized consistently, while non-native strains sometimes failed to colonize (aggregate colonization success, 100% (*n* = 93) versus 70% (*n* = 44); Fisher's exact test, *P* < 0.001) (Fig. 5a). The gut metabolomic comparison between successfully colonized versus MF bees was similar to our initial results. Carboxylic acids,

such as 3Hmg, malate, fumarate, succinate, isocitrate and glycerate were significantly less abundant in colonized guts versus MF controls across three or more of the *S. alvi* strains tested. The abundances of purine and amino acid precursors hypoxanthine and urea were also depleted in the guts of colonized bees, while anthranilate again accumulated in all colonized bees (Fig. 5b and Extended Data Fig. 8). These results show that the use of host-derived carboxylic acids is a conserved phenomenon in the genus *Snodgrassella* and facilitates gut colonization independent of bacterial crossfeeding or diet-derived nutrients.

## Discussion

Host-derived metabolites are gaining appreciation for their importance in facilitating extracellular microbial colonization across widely disparate animal models. In some cases, bacteria graze on the chitinous lining of the murine gut or of the light organ in squid[14–21]. In other instances, bacteria profit from small molecules secreted into the lumen. Commensal species can use lactate, 3-hydroxybutyrate and urea in the murine gut, while pathogenic *Salmonella* species use aspartate, malate, lactate or succinate to invade the gut[22–25]. Our study adds evidence that this phenomenon is conserved across widely disparate animal hosts and that these host-derived metabolites can represent the principal carbon source for certain gut bacteria facilitating colonization and growth, independent of the diet or crossfeeding.

We took advantage of the bee as a model host to drastically minimize confounding factors, colonizing bees with a single bacterium, *S. alvi*, while restricting the host to a sole dietary substrate that is undigestible for this bacterium. We then used $^{13}$C glucose labelling to show that organic acids measured in the gut are synthesized by the host from dietary sugars, which are then assimilated by *S. alvi* cells during gut colonization. The $^{13}$C enrichment of *S. alvi* was initially higher than in the surrounding host tissue but then dropped after the host diet was switched to $^{12}$C glucose. These dynamics are consistent

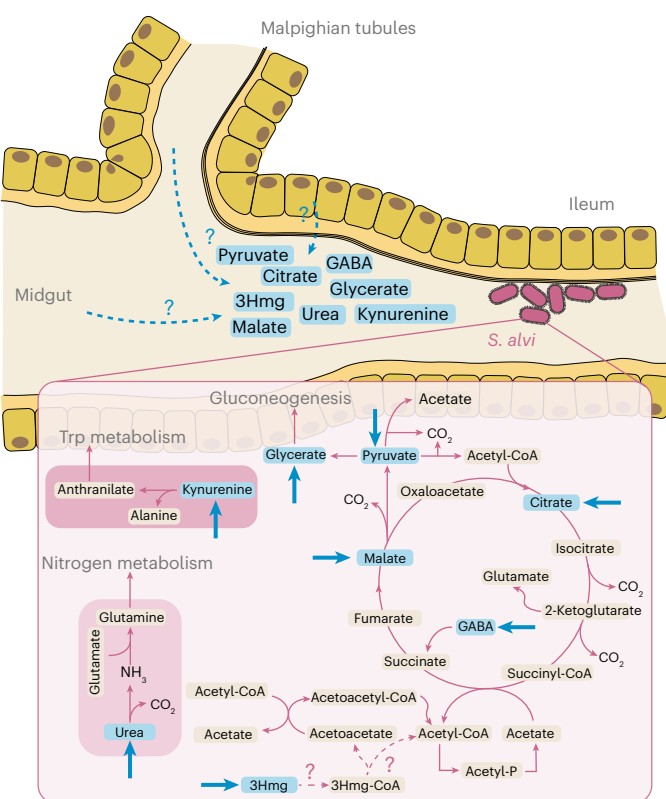

**Fig. 6 | Schematic model of *S. alvi* metabolism of host-derived compounds in the ileum.** Host metabolites (blue) enter the ileum through the epithelial cells or through the Malpighian tubules upstream of the ileum. *S. alvi* primarily uses the TCA cycle to generate energy, then synthesizing biomass through gluconeogenic reactions. 3Hmg provides a source of acetate to fuel the TCA cycle. However, the reactions synthesizing the first two steps (dotted pink arrows) have not been annotated in the genome. Host-synthesized urea is also a source of nitrogen and kynurenine is converted to anthranilate as part of tryptophan metabolism. Blue arrows indicate the entry of substrates provided by the host metabolism.

with the hypothesis that *S. alvi* uses host metabolites that are derived from simple carbohydrate catabolism in the bee diet, rather than from grazing on components of the host epithelium lining in the gut (Fig. 6). Finally, we demonstrated that these organic acids sustain growth of *S. alvi* in vitro.

Our results provide new context to the substantial body of research on bacterial colonization of the bee gut. The consistent depletion of host-derived acids linked to the TCA cycle complements findings that these metabolites are depleted in bees colonized with individual microbes including *S. alvi*, *Gilliamella*, *Lactobacillus* Firm-5 and *Bartonella apis*, as well as when colonized with a full microbial community[12,26,27]. Thus, foraging on host-derived compounds may be widespread in the native bee gut community. Yet, in the specific case of *S. alvi*, these host-derived nutrients seem to be key for colonization. This is supported by a previous study which found that all genes of the TCA cycle and many genes associated with organic acid and ketone body degradation provide a strong fitness advantage for *S. alvi* to grow in the bee gut[28].

Strikingly, two major metabolic functions conserved across all five strains of *S. alvi* (3Hmg consumption and anthranilate production) were associated with unique features in the metabolic pathways of *S. alvi*, providing evidence for adaptation to a specialized host niche. *S. alvi* possesses a non-canonical TCA cycle (Fig. 6); lacking a glyoxylate shunt it is unable to grow on acetate alone, but the substitution of acetate:succinate CoA-transferase for the canonical succinyl-CoA synthetase means that acetate, rather than acetyl-CoA, is a key driver of

the TCA cycle[29]. Interestingly, we found that 3Hmg consumption results in the production of acetate when *S. alvi* is grown on 3Hmg in vitro. Therefore, we postulate that host-derived 3Hmg enhances growth of *S. alvi* through the production of acetate, which in turn, increases the flux through the TCA cycle (Fig. 6). 3Hmg is an intermediate of the host's isoprenoid biosynthesis, leucine degradation and ketone body metabolism but the reason why this metabolite is released into the gut remains elusive.

The second unique metabolic feature of *S. alvi*, the production of anthranilate, probably depends on the enzymatic activity of a kynureninase gene identified in the genome of *S. alvi* converting host-derived kynurenine into anthranilate (Fig. 6). Intriguingly, this is the only gene annotated in the tryptophan degradation pathway of *S. alvi*; it is highly conserved in the entire genus and has probably been acquired by horizontal gene transfer, suggesting that it encodes a conserved function that is specific to the symbiosis between *Snodgrassella* and its bee hosts.

The release of relatively valuable metabolites into the gut could be a way for the host to control community assembly and facilitate the colonization of particularly beneficial gut symbionts[30]. While the full gut microbiota has been shown to carry out several important functions for the host, the specific role of *S. alvi* has remained elusive. Its metabolic activities may have positive effects on the host, as protection against pathogen invasion has been shown from kynurenine metabolism[31,32]. Kynurenine is important during the larval stage of bee development but it is also associated with neuronal defects (for example, hyperactivity and motor disfunction) in insects and vertebrates[33–35]. In contrast, anthranilate is an important precursor of the essential amino acid tryptophan and several beneficial neurotransmitters such as serotonin, tryptamine and various indole derivatives[36]. Intriguingly, indoles synthesized from tryptophan by *Lactobacillus* Firm-5 species in the gut were recently linked to enhanced memory and learning in honey bees[37]. Finally, we also note that the metabolism of urea by *S. alvi* could have a beneficial effect on the host. Urea constitutes a major waste product in mammals and insects. In turtle ants, ancient, specialized gut bacteria are able to recycle large amounts of nitrogen from urea into both essential and non-essential amino acids[38]. We found that urea is more abundant in the bee gut when nitrogen was absent from the diet, indicating higher metabolic turnover in the host, which *S. alvi* could alleviate through urea fixation.

Our findings fit within the context of several limitations. We examined a portion of the gut metabolome that is amenable to analysis via gas chromatography. Further relevant compounds may be detectable using complimentary methods. While we have shown that host-derived compounds can sustain growth of *S. alvi*, the tested conditions did not reflect the natural bee gut with a full microbial community and a metabolically rich diet that includes bee bread, pollen, honey and nectar. Therefore, we cannot exclude that dietary or microbially derived metabolites contribute to the growth of *S. alvi* in the native gut. In fact, we found in vivo evidence for crossfeeding of lactate and pyruvate from *Gilliamella*, as well as for pollen as an indirect source of key organic acids in the gut. However, these nutrient sources seem to play a minor role relative to the host, as the colonization levels of *S. alvi* did not differ significantly when pollen and/or *Gilliamella* were present. This is consistent with a previous study, which showed that the total abundance of *S. alvi*, in contrast to most other community members, does not change between nurse and forager bees that have different dietary preferences, nor does it change when pollen is removed from the diet of fully colonized bees under laboratory conditions[39].

How host-synthesized organic acids reach the gut lumen remains elusive (Fig. 6). While leakage from host cells across the epithelial barrier is possible, transport via the Malpighian tubules upstream of the ileum seems more likely, as they excrete nitrogenous waste and other metabolites into the gut while also regulating osmotic pressure in the haemolymph[40,41]. Future work may resolve this question through

careful dissection and metabolomic analysis of the Malpighian tubules and the ileum, possibly with the assistance of labelled compounds injected into the bee thorax.

## Methods

### Bacterial culturing

Extended Data Table 1 lists all strains used in this study[12,42,43]. Strains were isolated by directly plating gut homogenates on brain heart infusion agar (BHIA) or by first culturing them in Insectagro (15363611, Fisher Scientific) liquid media under microaerophilic conditions for 48 h before colonies were streaked on plates. Species identity was confirmed by 16S ribosomal RNA gene sequencing. Strains were grown on liquid Insectagro media for 18 h and then diluted in phosphate buffer saline (PBS) and 25% glycerol at optical density $OD_{600} = 1$. Solutions were kept frozen at −80 °C until just before colonization, when they were thawed on ice, diluted tenfold with PBS and mixed 1:1 with sterile sugar water (1 kg $l^{-1}$ of sucrose in water).

### Experimental colonization of gnotobiotic bees

All honey bees were collected from hives at the apiary of the University of Lausanne. A different hive was selected for each colonization replicate. MF bees were raised according to established protocols[44]. Briefly, frames containing capped brood were taken from the hive and washed quickly with a towel soaked in 3% bleach. Pupae of the appropriate age were then removed from the frames and placed into sterile emergence boxes containing 0.2 µm filter sterilized sucrose (50% w/w) sugar water. The boxes were placed in an incubator at 35 °C for 2 d, until the adult bees emerged. They were then randomly transferred to separate, sterile cages assembled from plastic drinking cups and Petri dishes.

Bees were colonized by hand with 5 µl of bacterial solution or corresponding blank control, after starving them of sugar water for 1–2 h, stunning them on ice for 5 min and transferring them with tweezers to individual microcentrifuge tubes that were cut to enable insertion of a pipette tip. Sugar water and, where specified, polyfloral pollen (Bircher Blütenprodukte) sterilized via electron beam irradiation (Studer Cables AG) were provided to cages ad libitum throughout each experiment. Mono- and co-colonization of bees with *S. alvi* wkB2 and the four *Gilliamella* strains were performed five times, using different hives for each treatment. Not all conditions were run in each experiment due to technical difficulties preparing the treatment groups and the GC–MS analysis was only carried out for a subset of the samples (Supplementary Table 1). The colonization experiment with the divergent *Snodgrassella* strains isolated from honey bees and bumble bees was carried out twice (Supplementary Table 2).

At the end of each experiment, bees were anaesthetized with $CO_2$, placed on ice and dissected with sterile forceps. Whole guts, minus the crop, were homogenized in a bead beater (FastPrep-24, MP) in tubes containing PBS and glass beads. The samples were each immediately divided into three aliquots. One was plated to c.f.u. on BHIA as well as for contaminants on nutrient agar (NA) and De Man, Rogosa and Sharpe agar (MRSA). Another aliquot of 500 µl was centrifuged for 15 min at 4 °C and 20,000g and the liquid supernatant snap frozen in liquid nitrogen and stored at −80 °C until metabolomic extraction. The third aliquot was frozen at −80 °C until DNA extraction.

### Validation of bee sterility

To rule out contamination in the guts of newly emerged bees, two bees from each emergence box in each experiment were sacrificed and 10 µl of their homogenized guts plated on BHIA (5% $CO_2$), NA (aerobic) and MRSA (anaerobic) at 35 °C to check for sterility before colonizing the remaining bees. In the event of contamination detected in newly emerged bees, all bees from that emergence box were discarded from the experiment. The same contamination checks were also performed on all analysed bees at the end of each experiment. Additional sterility checks were performed in the experiment where we compared gut

metabolite levels in newly emerged and 6-day-old MF bees. Here, we also plated eight homogenized guts on potato dextrose agar (aerobic), lysogeny broth agar (aerobic), tryptic soy agar (5% $CO_2$), Columbia agar + 5% sheep blood (CBA + 5%SB, 5% $CO_2$ and anaerobic) and tryptone yeast extract glucose agar (anaerobic).

### Quantification of pollen consumption

Both whole bees and dissected bee guts were weighed to assess treatment effects and quantify pollen consumption. Whole bee and wet gut weights were both significantly higher in bees provided with pollen than in those provided with sugar water only, while colonization did not significantly affect weight (Extended Data Fig. 4). Linear mixed-effects models fitted by maximum likelihood with nested, random cage effects: (bee weight, $n = 271$, pollen: $F_{(1,262)} = 176.1$, $P < 0.001$; *S. alvi*: $F_{(1,262)} = 0.1$, $P = 0.769$; *Gilliamella*: $F_{(1,262)} = 1.0$, $P = 0.311$; *S. alvi:Gilliamella*: $F_{(1,262)} = 0.7$, $P = .388$); (gut weight, $n = 271$, pollen: $F_{(1,262)} = 220.0$, $P < 0.001$; *S. alvi*: $F_{(1,262)} = 0.3$, $P = 0.595$; *Gilliamella*: $F_{(1,262)} = 0.02$, $P = 0.880$; *S. alvi:Gilliamella*: $F_{(1,262)} = 2.3$, $P = 0.131$). We also weighed filled pollen troughs at the beginning and end of the experiment and calculated the amount of pollen consumed per bee. Unsurprisingly, the mean difference in gut weights between dietary groups ($22.3 \pm 18.3$ mg per bee) closely matched the measured amount of pollen consumed by the bees across colonization treatments ($24.3 \pm 7.2$ mg per bee) (Supplementary Table 1). We then calculated the bee body weight (bee weight − gut weight) to determine if diet or colonization led to actual tissue weight gain in bees. While we found a slight increase in body weight from pollen (95% CI [0.69 mg, 7.51 mg]), we cannot rule out that this was due to variable amounts of pollen present in the crop, which was not removed with the hindgut (body weight, $n = 271$, pollen: $F_{(1,262)} = 5.2$, $P = 0.023$; *S. alvi*: $F_{(1,262)} = 1.4$, $P = 0.240$; *Gilliamella*: $F_{(1,262)} = 0.6$, $P = 0.442$; *S. alvi:Gilliamella*: $F_{(1,262)} = 3.8$, $P = 0.052$).

### Isotope tracing experiments

To assess the $^{13}C$ metabolite enrichment in the bee gut, MF bees were provided a 45:65 ratio of 99% uniformly labelled [U-$^{13}C_6$] glucose (CLM-1396, Cambridge Isotope Laboratory) and naturally abundant sucrose (S0389, Sigma Aldrich) in the first experiment. For the NanoSIMS time-course colonization experiment, newly emerged MF bees were divided into two cages and provided with pure 99% [U-$^{13}C_6$] glucose (treatment) or naturally abundant (that is ~98.9% $^{12}C$) glucose (control) in water for 5 days (pulse) before switching both cages to a naturally abundant glucose diet (chase). On the morning of the fourth day all bees were starved for 1.5 h and colonized with *S. alvi* wkB2. The $^{13}C$ glucose solution was replaced with a $^{12}C$ glucose solution 24 h later. Bees were collected at 0, 8, 24, 32, 48, 56, 72 h after inoculation. This overlap of colonization and $^{13}C$ glucose feeding was chosen to both maximize the number of bacterial cells in the transmission electron microscopy (TEM) sections and to reduce the dilution of $^{13}C$ label due to metabolic turnover. For each timepoint and both cages ($^{13}C$ and $^{12}C$ glucose treatment), the ileum of four bees were dissected. One gut was immediately preserved for electron microscopy and NanoSIMS, while the other three guts were homogenized and aliquoted for c.f.u. plating and metabolite extraction (Fig. 3). The control samples (from the cage provided with $^{12}C$ glucose) were used to determine the natural $^{13}C$ abundance in the bee tissue and *S. alvi* cells. Image metadata and raw NanoSIMS region of interest (ROI) values can be found in Supplementary Information.

### Quantification of bacterial loads via qPCR

DNA extraction was carried out by adding 250 µl of 2× CTAB (H6269, T9285, 81420, Sigma Aldrich) to frozen gut homogenates. The mixture was homogenized at 6 m $s^{-1}$ for 45 s in a Fast-Prep24TM5G homogenizer (MP Biomedicals) and centrifuged for 2 min at 2,000 r.p.m. DNA was extracted in two steps with Roti-Phenol (A156.3, Roth) and phenol:chloroform:isoamylalcohol (25:24:1), then eluted in 200 µl DNAse/RNAse

water and frozen. The qPCR was run using target-specific primers for *Apis mellifera* actin and specific 16S rRNA gene primers for *S. alvi* and *Gilliamella* spp. as well as 16S rRNA gene universal primers used in ref. 12. However, amplification of chloroplast DNA was also observed in samples containing pollen, as shown in previous studies[45].

The qPCR reactions were run on a QuantStudio 3 (Applied Biosystems) and standard curves were performed using the respective amplicons on a plasmid as previously described[12].

In addition, 18S rRNA gene amplification with universal primers were run to assess fungal loads (F: TATGCCGACTAGGGATCGGG, R: CTG-GACCTGGTGAGTTTCCC)[46]. The qPCR reactions were run on 384-well plate QuantStudio 5 (Applied Biosystems). The corresponding standard curve for absolute quantification was performed using the 199 base pair amplicon as opposed to a plasmid containing the amplicon. Data were extracted and analysed as previously described[12].

## Metabolite extractions

Frozen gut samples were divided and extracted via two established methods for SCFAs and soluble metabolites. Before soluble metabolite extraction from pure pollen grains, an equivalent amount to what was digested per bee was mixed with PBS. To approximate physiochemical degradation of the pollen in the bee gut, the samples were bead beaten and incubated at 35 °C for 18 h.

SCFAs were extracted with 750 µl of diethyl ether from 75 µl of gut supernatant that was first acidified with 5 µl of 11% HCl. Isovalerate (200 µM) (129542, Sigma Aldrich) was used as an internal standard. The solvents and resulting two-phase mixture were kept cold at −20 °C and vortexed 3× for 30 s. The samples were then centrifuged for 3 min at 4 °C and 13,000$g$ and 80 µl of the ether phase was removed to a glass GC vial (5188-6591, Agilent). The sample was derivatized with 20 µl of MTBSTFA + 1% TBDMS (00942, Sigma Aldrich) for 1 h at 30 °C (ref. 47).

Soluble metabolites were extracted via a modified Bligh and Dyer protocol[48]. A (5:2:1) mixture of methanol (1.06009, VWR) and chloroform (1.02444, VWR) and double-distilled water was chilled to −20 °C and 800 µl added to 100 µl of gut homogenate. A mixture of L-norleucine (N8513, Sigma Aldrich), DL-norvaline (53721, Sigma Aldrich) and $^{13}$C glucose was used as internal standards. The samples were vortexed three times for 30 s and extracted at −20 °C for 90 min. The samples were then centrifuged at 13,300$g$ and 4 °C for 5 min. Liquid supernatant was transferred to a new tube and 400 µl of cold chloroform: methanol (1:1) was added to the insoluble material. The samples were again vortexed three times, extracted for 30 min at −20 °C, centrifuged and the two supernatants were combined. Phase separation was achieved by adding 200 µl of water and centrifuging at the same conditions. The upper phase was then dried overnight in a speed vacuum. The sample was derivatized with 50 µl of 20 mg ml$^{-1}$ methoxyamine hydrochloride in pyridine (226904, Sigma Aldrich), for 90 min at 33 °C followed by silylation with 50 µl of MSTFA (69478, Sigma Aldrich) for 120 min at 45 °C.

## GC−MS analysis

Samples were analysed on an Agilent 8890/5977B series GC-MSD equipped with an autosampler that injected 1 µl of sample onto a VF-5MS (30 m × 0.25 mm × 0.25 µm) column. The SCFA samples were injected with a split ratio of 25:1, helium flow rate of 1 ml min$^{-1}$ and inlet temperature of 230 °C. The oven was held for 2 min at 50 °C, raised at 25 °C min$^{-1}$ to 175 °C and then raised at 30 °C min$^{-1}$ to 280 °C and held for 3.5 min. The MSD was run in SIM mode with three target ions for each compound. The soluble metabolite samples were injected with a split ratio of 15:1, helium flow rate of 1 ml min$^{-1}$ and inlet temperature of 280 °C. The oven was held for 2 min at 125 °C, raised at 3 °C min$^{-1}$ to 150 °C, 5 °C min$^{-1}$ to 225 °C and 15 °C min$^{-1}$ to 300 °C and held for 1.3 min. The MSD was run in scan mode from 50 to 500 Da at a frequency of 3.2 scan s$^{-1}$.

## Metabolomics data analysis

Mass spectrometry features were identified by spectral matching to the NIST17 mass spectra library or to inhouse analytical standards. Features were then extracted using the MassHunter Quantitative Analysis software (Agilent). Data normalization and analysis was performed using custom R Studio (4.0.3) scripts (see Data availability). Absolute metabolite quantification was performed by fitting normalized response values to calibration curves of analytical standards. The effect of diet and colonization with *S. alvi* and *Gilliamella* were assessed by fitting $z$-score normalized metabolite abundances to a linear mixed-effects model fitted by restricted maximum likelihood (REML) using the lmm2met package in R with diet, *S. alvi* and *Gilliamella* as fixed effect factors and experimental replicate (bee colony) as a random effect[49]. The significance of each fixed effect was then adjusted by the Benjamini−Hochberg (BH) procedure to control for false discovery. Mass isotopologues were adjusted to account for natural isotope abundances following published procedures[50,51].

## TEM section preparation and NanoSIMS analysis

Bee guts were dissected, the ileum was cut away from the rest of the gut and fixed in glutaraldehyde solution (EMS) 2.5% in phosphate buffer (PB, 0.1 M pH 7.4) (Sigma) for 2 h at room temperature (RT) and kept at 4 °C until sample preparation. Ileum pieces were rinsed three times for 5 min in PB buffer and then postfixed with a fresh mixture of osmium tetroxide 1% (EMS) with 1.5% potassium ferrocyanide (Sigma) in PB buffer for 2 h at RT. The samples were then washed three times in distilled water and dehydrated in acetone solution (Sigma) at graded concentrations (30%−40 min; 70%−40 min; 100%−1 h; 100%−2 h). Dehydration was followed by infiltration in Epon resin (Sigma) at graded concentrations (Epon 1/3 acetone−2 h; Epon 3/1 acetone−2 h, Epon 1/1−4 h; Epon 1/1−12 h) and finally polymerized for 48 h at 60 °C in an oven. Thin sections of 100 nm were cut on a Leica Ultracut (Leica Mikrosysteme GmbH) and picked up on copper slot grid 2 × 1 mm (EMS) coated with a polyetherimide film (Sigma). Sections were poststained with uranyl acetate (Sigma) 2% in H$_2$O for 10 min, rinsed several times with H$_2$O followed by Reynolds lead citrate for 10 min and rinsed several times with H$_2$O.

Large montages with a pixel size of 6.9 nm covering areas of around 120 × 120 µm were taken with a transmission electron microscope (TEM) Philips CM100 (Thermo Fisher Scientific) at an acceleration voltage of 80 kV with a TVIPS TemCam-F416 digital camera (TVIPS GmbH) and the alignment was performed using Blendmont command-line program from the IMOD software[52].

Four holes were made using the electron beam at the four corners of the montage to localize the area of interest for NanoSIMS imaging. Then the slot grid was deposited on 1 µl of distilled water on a 10 mm round coverslip and left to dry for 10 min. The slot grid was detached, letting the section attach to the coverslip.

After being coated with a 15 nm gold layer to evacuate a potential build-up of a charge during the NanoSIMS analyses, the samples were analysed by a NanoSIMS 50 l, software v.4.5 (CAMECA). A primary beam of 16 keV Cs+ ions was focused to a spot size of about 100 nm on the sample surface, causing the sputtering of atoms and small molecules from the first 5−10 nm of the sample. Negative secondary ions were directed and focalized into a single beam before reaching the magnetic sector of the mass spectrometer where the ions were separated. The species of interest ($^{12}$C$^{14}$N- and $^{13}$C$^{14}$N- for this study) were then measured simultaneously using electron multipliers. To resolve the interference between $^{13}$C$^{14}$N- and $^{12}$C$^{15}$N-, two slits (ES and AS) were used to achieve a mass resolution higher than 10,000 (Cameca definition)[53].

In the imaging mode, selected areas of 25 × 25 µm$^2$ were implanted using a strong beam (using diaphragm D1-1 (diameter 750 µm) and a current of 16 pA) to remove the gold coating and reach stable emission conditions. After optimization of the extraction of the secondary ions, the area was imaged by scanning with a smaller

**Article** https://doi.org/10.1038/s41564-023-01572-y

beam (~100 nm, using diaphragm D1-5 (diameter 100 μm) and a current of 0.45 pA) with a 256 × 256 pixel resolution and a dwell time of 5 ms per pixel. For each image, ten layers were acquired. The image processing was performed using L'image software version 10-15-2021 (L. Nittler, Carnegie Institution of Washington). The ten layers were aligned and stacked, 44 ms deadtime correction was applied. Regions of interests (ROI) were manually drawn around bacterial cells, gut epithelium layer and bulk host cells. The resulting data are reported in Supplementary Table 3. The at.% enrichment was calculated in R using the data.table package. The at.% enrichment formula: $^{13}C$ at.% = $^{14}N$ $^{13}C/(^{14}N$ $^{13}C + ^{14}N$ $^{12}C) \times 100$ (Supplementary Fig. 1).

### In vitro growth of *S. alvi*

*S. alvi* was cultured in Bee9 media, an M9-derived minimal defined media containing essential vitamins and mixtures of organic acids (Supplementary Information). Growth assays were performed in 96-well plates (Corning costar) at 35 °C, 5% $CO_2$, with continuous orbital shaking (H1 Synergy, Biotek). $OD_{600}$ measurements were taken every 30 min for 20 h. All conditions were independently replicated three times. For metabolomics measurements, plates were immediately placed on ice at the end of each experiment and liquid was transferred to microcentrifuge tubes. The tubes were then centrifuged at 13,300g and 4 °C for 30 s and the liquid supernatant was then snap frozen in liquid nitrogen and stored at −80 °C for metabolomic analysis.

### Phylogenetic analysis of the kynureninase gene family

The kynureninase protein sequence of *S. alvi* wkB2 was retrieved from NCBI (WP 025330329.1) and blasted (NCBI, BlastP) against the nr database. On the basis of the BlastP hits, 227 divergent amino acid sequences were exported in fasta format and aligned using MUSCLE[54] with standard parameters. Tree inference and bootstrap validation were performed using IQtree with the following command: iqtree-omp -s kyn_sequences_muscle.aln -nt 6 -m TEST -bb 1000 (ref. 55). The tree topology was visualized using the ITOL online viewer (https://itol.embl.de/).

### Statistics and reproducibility

Bees, grouped by hive, were randomly assigned to treatment conditions. Data processing and analysis, that is GC–MS or qPCR, of bee material and in vitro samples was performed randomly across treatment groups and experimental replicates to prevent confounding of treatment with batch effects. The investigators were not blinded to allocation during experiments and outcome assessment. No statistical methods were used to predetermine sample sizes but our sample sizes are similar to those reported in previous publications[12,26]. One experimental replicate (Fig. 1) was discarded entirely from the analysis due to detected contamination in the bee emergence boxes before treatment. For quantifying bacterial abundance with qPCR, samples with s.d. >0.7 were excluded from the analysis. Similarly, samples were removed from the metabolomics dataset when internal standards greater than or less than 2 × s.d. from the median of that batch (Fig. 1, Extended Data Figs. 2 and Fig. 3: n = 6; Fig. 5 and Extended Data Fig. 8: n = 3).

### Reporting summary

Further information on research design is available in the Nature Portfolio Reporting Summary linked to this article.

## Data availability

The complete set of raw data have been deposited on Zenodo at https://doi.org/10.5281/zenodo.10066636. Source data are provided with this paper.

## Code availability

Computational code used to analyse data and generate figures are also available in the Zenodo repository at https://doi.org/10.5281/zenodo.10066636.

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

## Acknowledgements

We would like to thank S. Moriano-Gutierrez and G. Santos-Matos for providing helpful comments on the manuscript. This work was funded by the University of Lausanne, an ERC Starting Grant (MicroBeeOme), the NCCR Microbiomes, a National Centre of Competence in Research, funded by the Swiss National Science Foundation (grant no. 180575) and a Swiss National Science Foundation project grant (grant no. 179487) to P.E.

## Author contributions

A.Q., Y.E. and P.E. conceived the project and designed the experiments. A.Q. and Y.E. performed the experiments. A.Q. analysed the GC–MS results, while Y.E. analysed the qPCR and NanoSIMS results. N.N. assisted with bee experiments, metabolite and DNA extractions. A.M.N. performed preliminary in vitro growth experiments. J.D. performed the tissue sectioning and TEM imaging supervised by C.G. S.E. performed the NanoSIMS imaging and instructed on the data analysis, supervised by A.M., who also advised on the design of the NanoSIMS experiment. A.Q., Y.E. and P.E. wrote the manuscript, with contributions from S.E. and A.M. All authors reviewed the manuscript and provided feedback. As equally contributing first authors, A.Q. and Y.E. may each list their name first when referencing this work.

## Competing interests

The authors declare no competing interests.

## Additional information

**Extended data** is available for this paper at

**Supplementary information** The online version contains supplementary
material available at https://doi.org/10.1038/s41564-023-01572-y.

**Correspondence and requests for materials** should be addressed to
Philipp Engel.

**Peer review information** *Nature Microbiology* thanks Manuel Liebeke
and the other, anonymous, reviewer(s) for their contribution to the
peer review of this work. Peer reviewer reports are available.

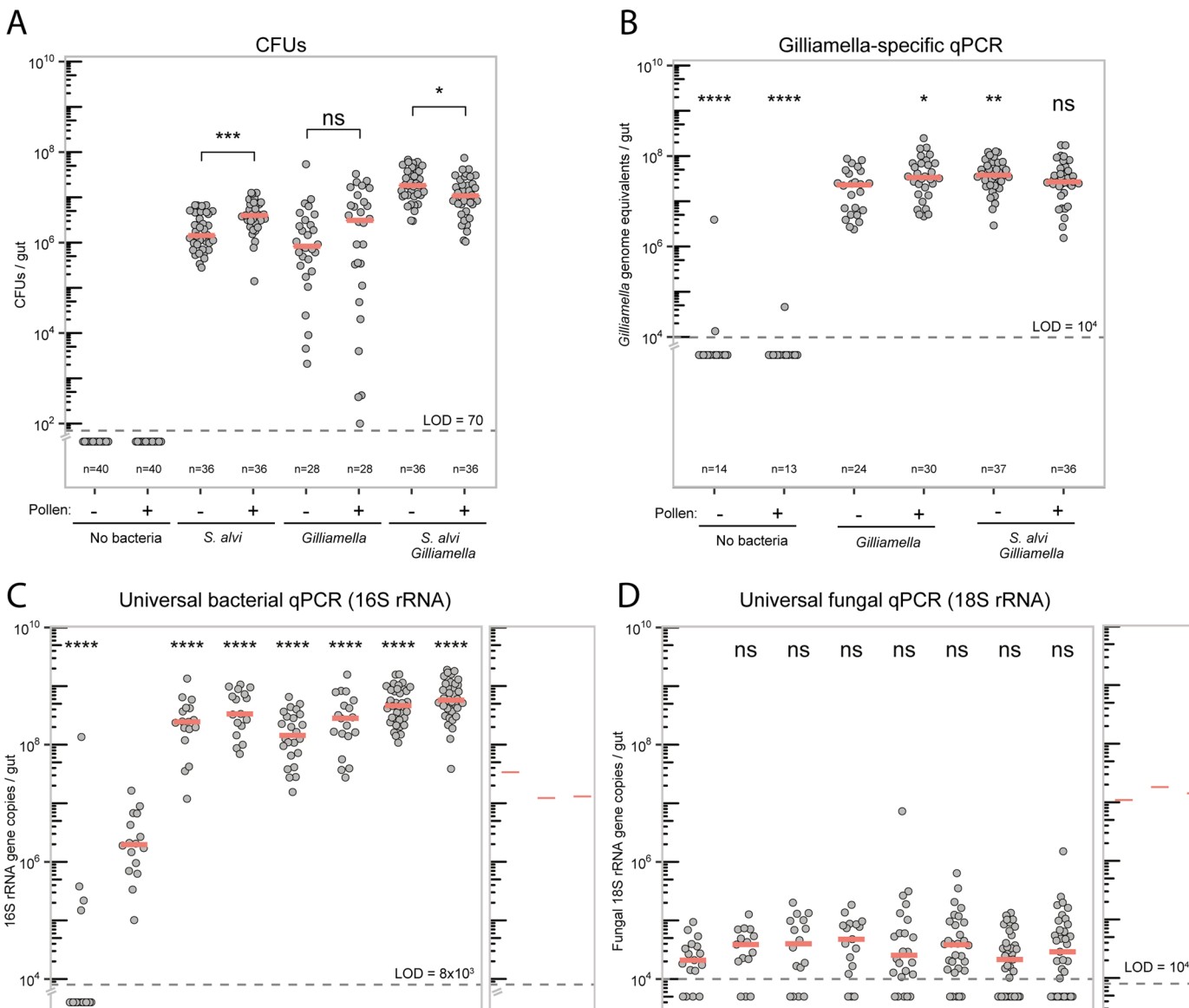

**Extended Data Fig. 1 | Total number of bacteria and fungi and colonization levels of *Gilliamella* per bee gut. (A)** Total number of bacteria (both *S. alvi* and *Gilliamella*) was determined by plating homogenized guts on BHIA and counting CFUs. Horizontal dotted line represents LOD < 70. No colonies were observed in samples plotted below the LOD line. Significant differences in CFUs between dietary conditions was determined with a two-sided Wilcoxon rank sum test (*S. alvi*: ***$P$ = 3.9x10$^{-4}$; *S. alvi* + *Gilliamella*: *$P$ = 0.031) NS, non-significant. **(B)** Bacterial cells (number of bacterial genome equivalents) per bee gut determined by qPCR using genus-specific *Gilliamella* 16 S rRNA gene primers. The horizontal dotted line represents qPCR LOD < 10$^4$. A two-sided Wilcoxon rank sum test was used with *Gilliamella* (Pollen -) as reference group, NS, non-significant; (No bacteria: ****$P$ = 4.43x10$^{-7}$, No bacteria (Pollen) ****$P$ = 4.25x10$^{-7}$, *Gilliamella* (Pollen): *$P$ = 3.7x10$^{-2}$, *S. alvi* + *Gilliamella*: **$P$ = 6x10$^{-3}$). **(C)** Number of 16 S rRNA gene copies per bee gut and in three different quantities of sterilized bee pollen as determined by qPCR using universal 16 S rRNA gene primers. The

horizontal dotted line represents qPCR LOD < 8x10$^3$. Wilcoxon rank sum test was used with MF (Pollen -) as reference group, (No bacteria: ****$P$ = 3.16x10$^{-5}$, *S. alvi*: ****$P$ = 1.71x10$^{-9}$, *S. alvi* (Pollen) ****$P$ = 1.71x10$^{-9}$, *Gilliamella*: ****$P$ = 3.18x10$^{-11}$, *Gilliamella* (Pollen): ****$P$ = 4.93x10$^{-10}$, *S. alvi* + *Gilliamella*: ****$P$ = 1.35x10$^{-13}$, *S. alvi* + *Gilliamella* (Pollen): ****$P$ = 1.93x10$^{-13}$ NS, non-significant). The number of 16 S rRNA gene copies detected in DNA extracted from different quantities of sterile pollen used in the bee diet is shown in a separate plot. The high background signal in samples containing pollen comes from the amplification of chloroplast DNA as shown in previous studies[45]. **(D)** Number of 18 S rRNA gene copies per bee gut and in three different quantities of sterilized bee pollen as determined by qPCR using universal 18 S rRNA gene primers. The horizontal dotted line represents qPCR LOD < 10$^4$. Wilcoxon rank sum test was done with MF (Pollen -) as reference group, NS, non-significant. In all plots, each dot represents a single bee gut and red bars represent the median value for each group. The absence or presence of pollen in the diet is indicated by (-/+).

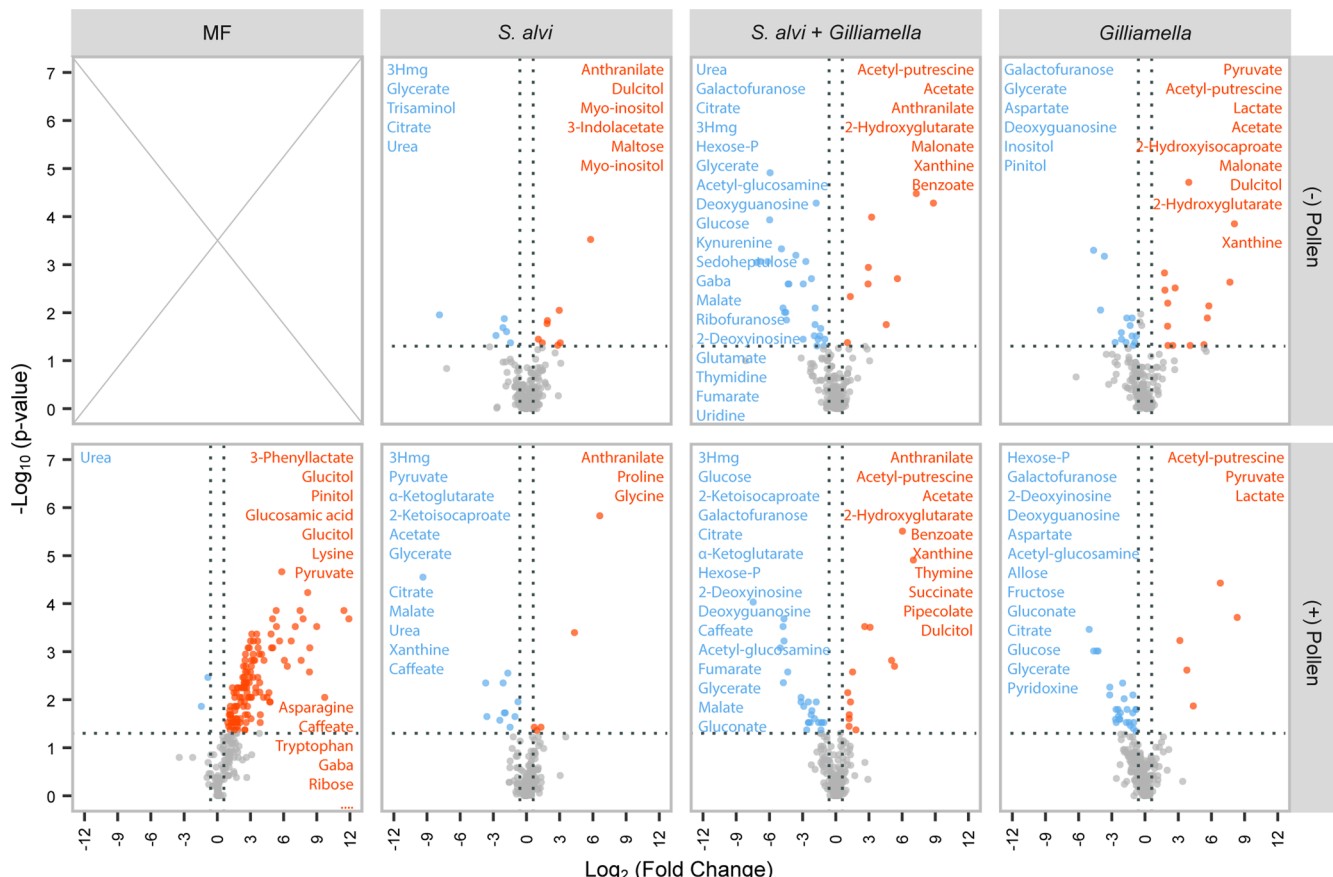

**Extended Data Fig. 2 | Volcano plots highlighting gut metabolite abundance changes between colonized and microbiota-free (MF) bees with and without pollen.** Significantly differentially abundant metabolites between mono- and cocolonized *S. alvi* and *Gilliamella* vs MF bees, as well as between MF (Pollen +) versus MF (Pollen -) bees (bottom left). Significantly depleted and produced metabolites are shown in blue and orange, respectively. Annotations are listed in order of significance for identified metabolites, with isomers and other duplications removed. The full set of significantly more abundant metabolites in the first panel can be found in the supporting code. Adjusted significance values were calculated using a two-sided Wilcoxon rank sum test with BH correction, ((pollen ±): MF: n = 12/10; *S. alvi*: n = 11/12; *S. alvi* + *Gilliamella*: n = 13/14; *Gilliamella*: n = 10/13).

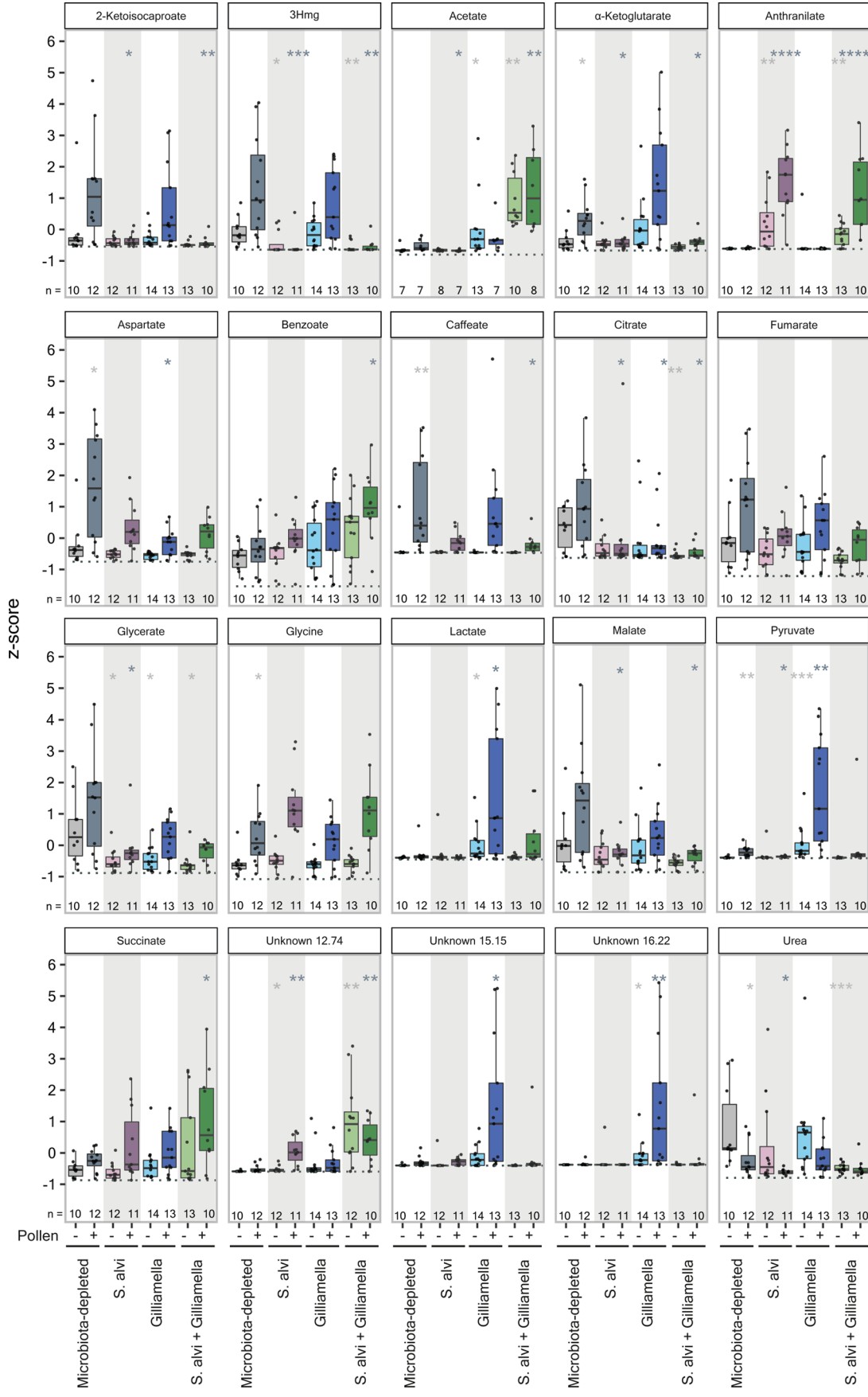

**Extended Data Fig. 3 | See next page for caption.**

**Extended Data Fig. 3 | Z-score normalized metabolite abundances of compounds significantly affected by *S. alvi* colonization.** Box plots and individual data points display values across all 8 experimental conditions. Only metabolites in which a mixed linear model showed a significant effect of *S. alvi* colonization are plotted. Multiple types of metabolic interactions are shown. For example, metabolite depletion by *S. alvi* exclusively (2-ketoisocaproate, 3Hmg, caffeate, urea) or by both species (aspartate, citrate, glycerate), metabolite production by *S. alvi* exclusively (anthranilate, unknown 12.74), crossfeeding from *Gilliamella* to *S. alvi* (pyruvate, lactate), synergistic metabolite depletion (fumarate, malate) and production (acetate, benzoate). The dark and light grey asterisks show conditions significantly different from MF bees fed sugar water and pollen or only sugar water (also indicated by ± sign), calculated using a two-sided Wilcoxon rank sum test with BH correction, *$P < 0.05$, ** $P < 0.01$, *** $P < 0.001$, **** $P < 0.0001$. ((Pollen ±): P values are shown in Supplementary Table 3. No bacteria: n = 12/10; *S. alvi*: n = 11/12; *S. alvi* + *Gilliamella*: n = 13/14; *Gilliamella*: n = 10/13). The median and the inter-quartile range (IQR) are depicted by the box, with whiskers extending to the furthest data points up to 1.5*IQR. The dotted line represents the transformed LOD for each metabolite.

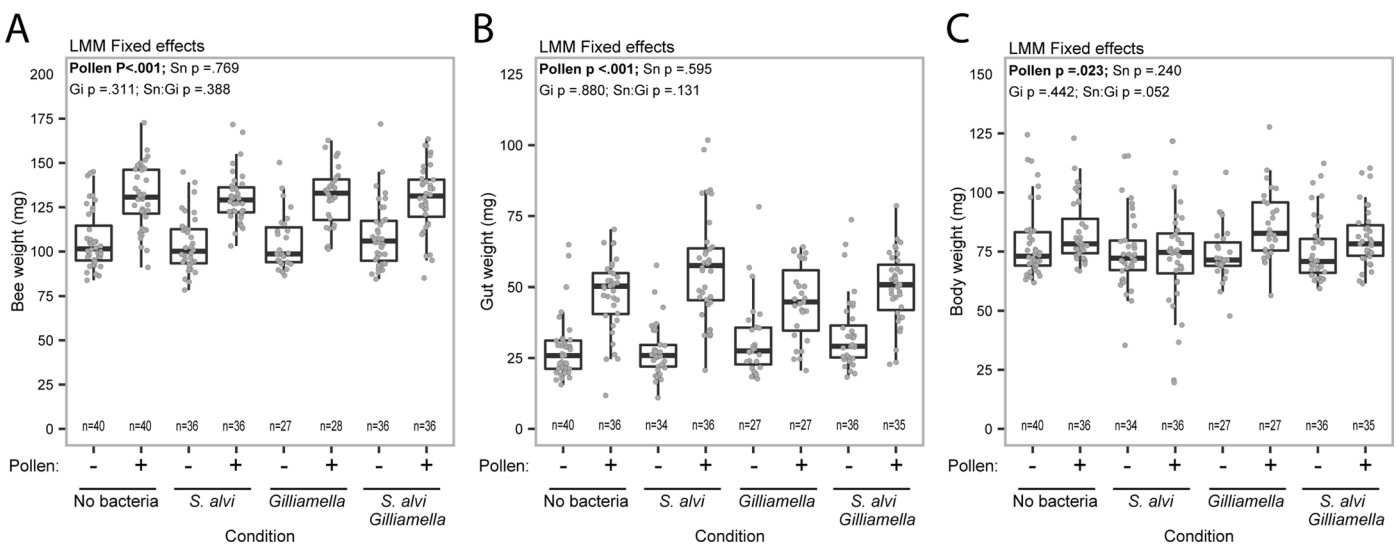

**Extended Data Fig. 4 | Pollen consumption but not colonization increases bee gut weight.** Increased weights with pollen in the diet of whole bees (**A**) (95% CI [22.1 mg, 29.7 mg]) and dissected bee Guts (**B**) (95% CI [19.0 mg, 24.8 mg]) are significantly higher, but no effect was measured when colonized with *S. alvi* and/ or *Gilliamella* [Bee weight, n = 271, Diet: $F_{(1,262)}$ = 176.1, $P < .0001$; *S. alvi*: $F_{(1,262)}$ = 0.1, $P = .769$; *Gilliamella*: $F_{(1,262)}$ = 1.0, $P = .311$; *S. alvi:Gilliamella*: $F_{(1,262)}$ = 0.7, $P = .388$], [Gut weight, n = 271, Diet: $F_{(1,262)}$ = 220.0, $P < .0001$; *S. alvi*: $F_{(1,262)}$ = 0.3, $P = .595$; *Gilliamella*: $F_{(1,262)}$ = 0.02, $P = .880$; *S. alvi:Gilliamella*: $F_{(1,262)}$ = 2.3, $P = .131$]. (**C**) The body weight, or difference between bee and gut weights, was also significantly, albeit only slightly (95% CI [0.7 mg, 7.5 mg]) higher across colonization groups for bees fed a pollen diet [Body weight, n = 271, Pollen: $F_{(1,262)}$ = 5.2, $P = .023$; *S. alvi*: $F_{(1,262)}$ = 1.4, $P = .240$; *Gilliamella*: $F_{(1,262)}$ = 0.6, $P = .442$; *S. alvi:Gilliamella*: $F_{(1,262)}$ = 3.8, $P = .052$]. The p values (ANOVA) are shown for the fixed effect terms in the linear mixed models fitted by ML with nested, random cage effects. The median and the IQR are depicted by the box, with whiskers extending to the furthest data points up to 1.5 xIQR. ((Pollen ±): No bacteria: n = 40/40; *S. alvi*: n = 36/36; *S. alvi + Gilliamella*: n = 36/36; *Gilliamella*: n = 28/27).

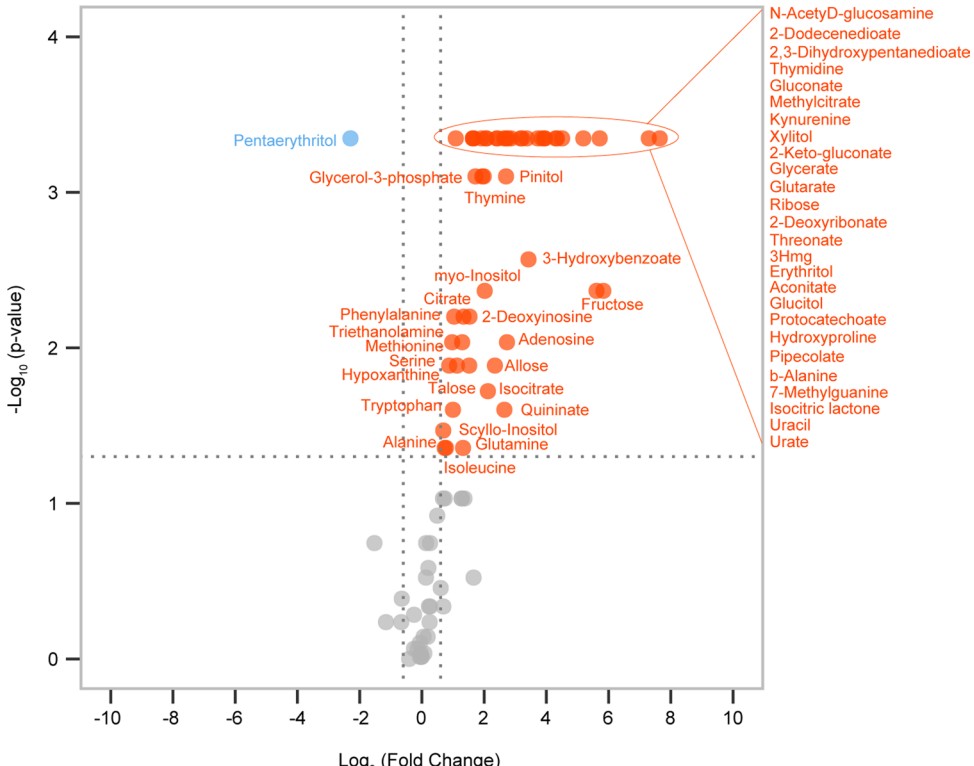

**Extended Data Fig. 5 | Volcano plot highlighting gut metabolic changes between Day 0 and Day 6 after emergence in MF bees without pollen.** Many metabolites are more abundant in the gut on Day 6 than on Day 0 after emergence. Significantly less and more abundant metabolites on Day 6 are shown and annotated in blue and orange, respectively, with isomers and other duplications removed. Adjusted significance values were calculated using two-sided Wilcoxon rank sum test with BH correction, using the day zero conditions as a reference (Day 0: n = 8; Day 6: n = 8).

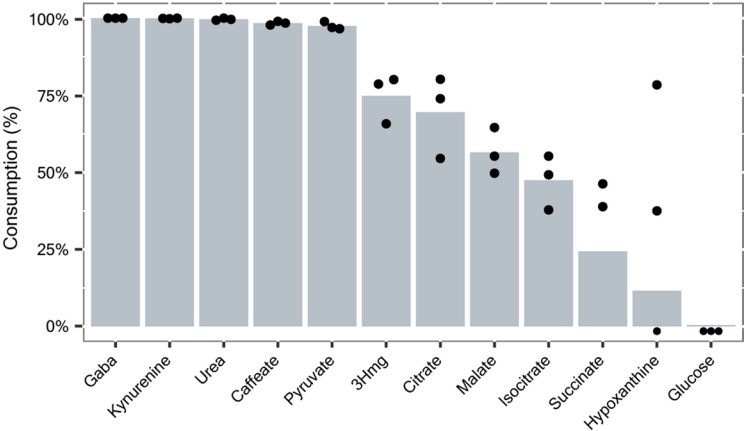

**Extended Data Fig. 6 | *In vitro* consumption of carbon substrates.** The percentage of each carbon source consumed by the end of the *in vitro* growth experiment is shown for each tested carbon source. Data was taken from the condition in which all carbon sources were added to Bee9 media in equimolar amounts. Bars represent the average value (n = 3) of biological replicates (black dots).

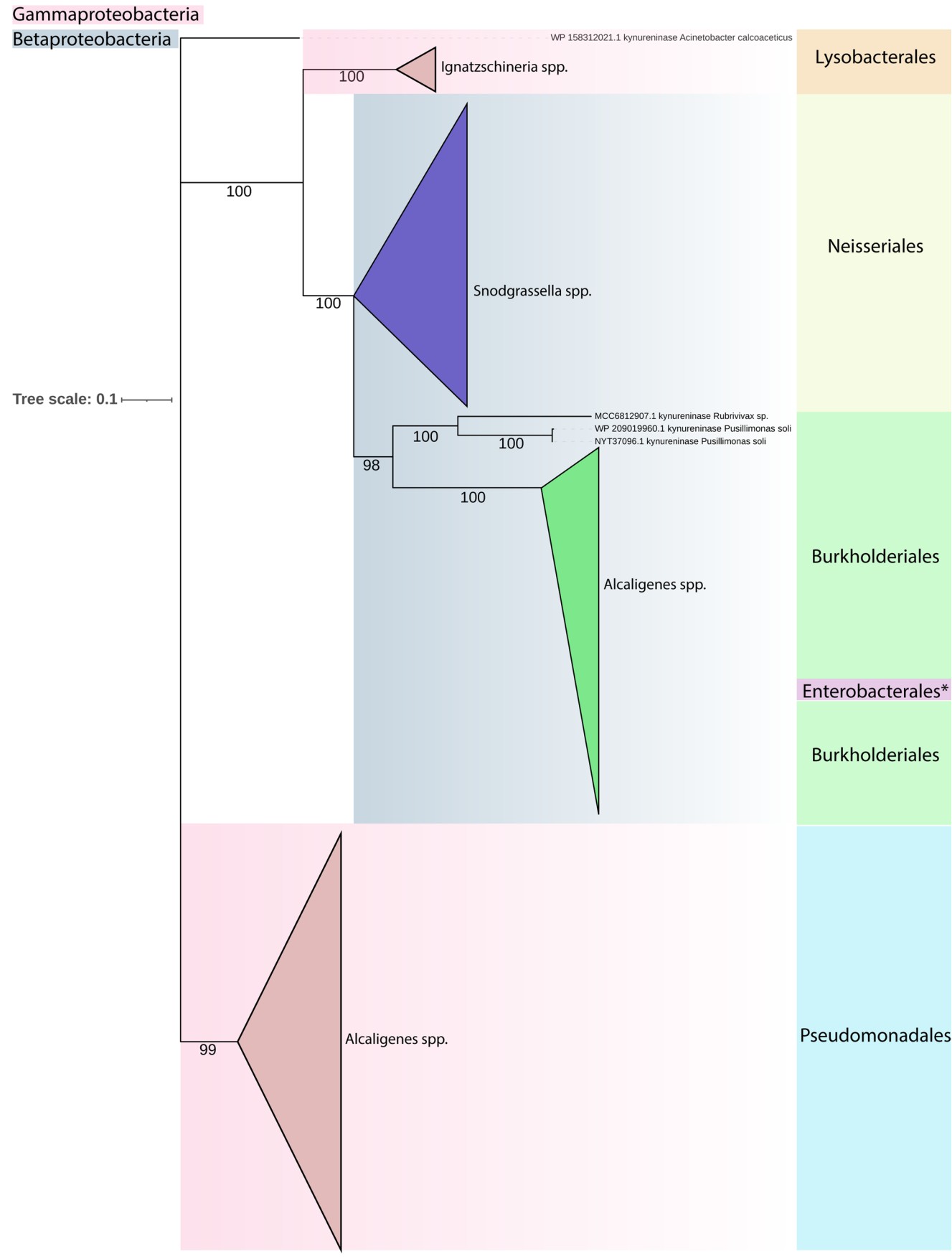

Gammaproteobacteria

Betaproteobacteria

WP 158312021.1 kynureninase Acinetobacter calcoaceticus

Lysobacterales

Ignatzschineria spp.

Neisseriales

Snodgrassella spp.

Tree scale: 0.1

MCC6812907.1 kynureninase Rubrivivax sp.
WP 209019960.1 kynureninase Pusillimonas soli
NYT37096.1 kynureninase Pusillimonas soli

Alcaligenes spp.

Burkholderiales

Enterobacterales*

Burkholderiales

Alcaligenes spp.

Pseudomonadales

**Extended Data Fig. 7 | See next page for caption.**

**Extended Data Fig. 7 | Phylogenetic tree of the kynureninase gene family found in *S. alvi*.** The amino acid sequences of 227 homologues of the kynureninase gene identified in *S. alvi* wkb2 were retrieved from NCBI. The tree was inferred with Iqtree after aligning the sequences with MUSCLE (see methods). Tips are collapsed for clades that contain sequences of the same genus. Coloured boxes on the right indicate the bacterial family. Bootstrap values are indicated before every node. Gammaproteobacteria are highlighted in pink and Betaproteobacteria in blue. Asterisk indicates a sequence (MBY6345856.1) from *Providencia rettgeri*, a Gammaproteobacterium, which clusters together with homologues of the kynureninase of Betaproteopbacteria.

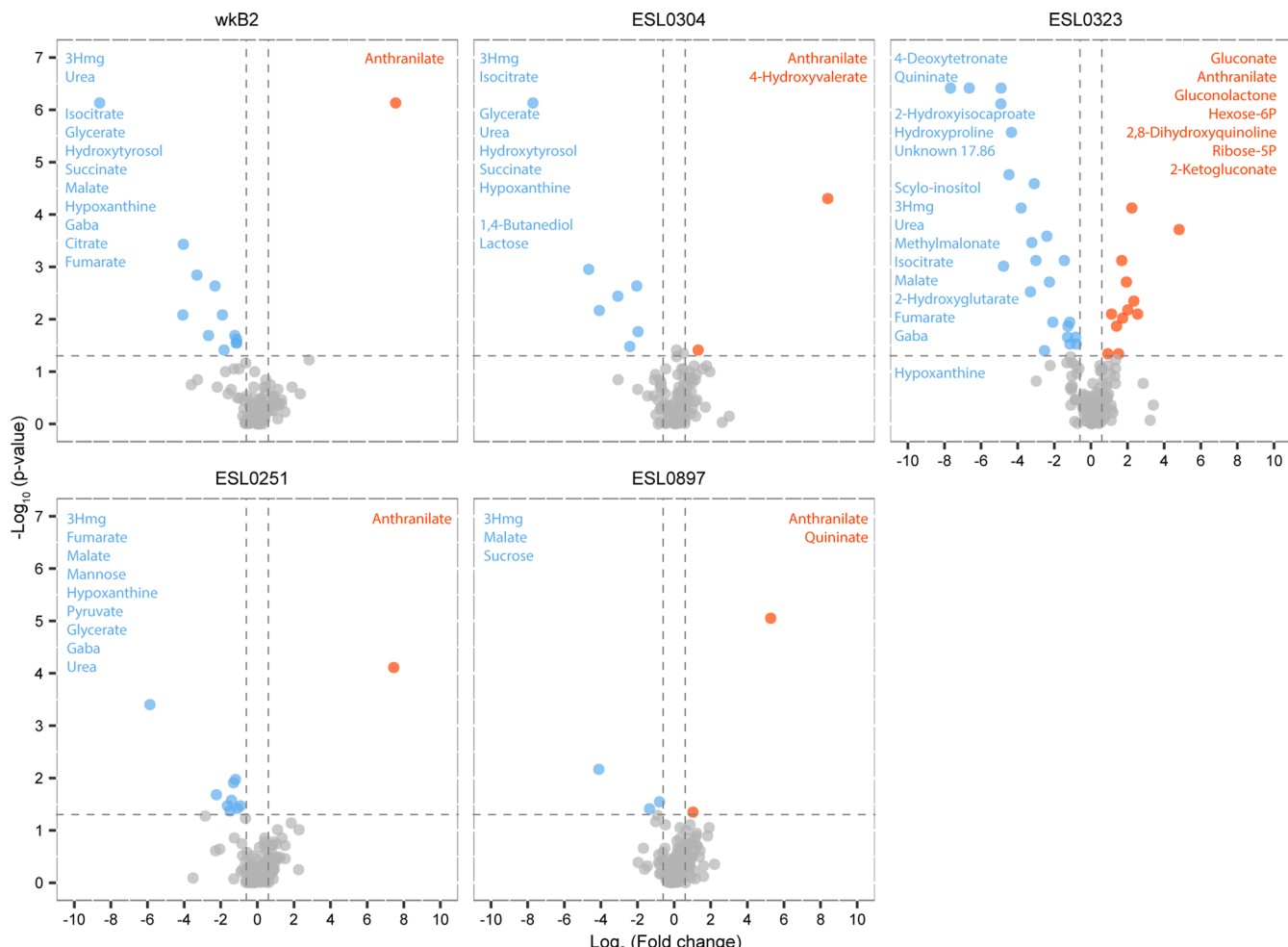

**Extended Data Fig. 8 | Gut metabolite changes in bees monocolonized with divergent *Snodgrassella* strains relative to microbiota-free (MF) bees.** Volcano plot of gut metabolites showing the similarities in metabolite changes from colonized vs MF bees across strains. Top: Strains native to *A. mellifera*; Bottom: Strains native to *Bombus sp*. Significantly depleted and produced metabolites are shown in blue and orange, respectively. Adjusted significance values were calculated using a two-sided Wilcoxon rank sum test with BH correction.

**Extended Data Table 1 | Bacterial strains used in this study**

| Bacterial strain | # 16S rRNA copies | Culturing condition | Strain source | Place of origin |
|---|---|---|---|---|
| Gilliamella apicola wkB1[T] | 4 | BHIA, 35°C, microaerophilic | (Kwong & Moran, 2013) | New Haven, USA |
| Gilliamella apis ESL0169 | 4 | BHIA, 35°C, microaerophilic | (Kešnerová et al., 2017) | Lausanne, Switzerland |
| Gilliamella sp. ESL0182 | 4 | BHIA, 35°C, microaerophilic | (Ellegaard & Engel, 2018) | Lausanne, Switzerland |
| Gilliamella apis ESL0297 | 4 | BHIA, 35°C, microaerophilic | This study | Lausanne, Switzerland |
| Snodgrassella alvi wkB2[T] | 4 | TSA, 35°C, microaerophilic | (Kwong & Moran, 2013) | New Haven, USA |
| Snodgrassella alvi ESL0304 | 4 | TSA, 35°C, microaerophilic | This study | Lausanne, Switzerland |
| Snodgrassella alvi ESL0323 | 4 | TSA, 35°C, microaerophilic | This study | Lausanne, Switzerland |
| Snodgrassella alvi ESL0251 | 4 | TSA, 35°C, microaerophilic | This study | Lausanne, Switzerland |
| Snodgrassella alvi ESL0897 | 4 | TSA, 35°C, microaerophilic | This study | Lausanne, Switzerland |

All strains used in this study, their 16S rRNA copy number, culturing conditions and original source are listed here. refs. 12,41,42

# Reporting Summary

## Statistics

For all statistical analyses, confirm that the following items are present in the figure legend, table legend, main text, or Methods section.

| n/a | Confirmed | |
|---|---|---|
| ☐ | ☒ | The exact sample size ($n$) for each experimental group/condition, given as a discrete number and unit of measurement |
| ☐ | ☒ | A statement on whether measurements were taken from distinct samples or whether the same sample was measured repeatedly |
| ☐ | ☒ | The statistical test(s) used AND whether they are one- or two-sided<br>*Only common tests should be described solely by name; describe more complex techniques in the Methods section.* |
| ☒ | ☐ | A description of all covariates tested |
| ☐ | ☒ | A description of any assumptions or corrections, such as tests of normality and adjustment for multiple comparisons |
| ☐ | ☒ | A full description of the statistical parameters including central tendency (e.g. means) or other basic estimates (e.g. regression coefficient) AND variation (e.g. standard deviation) or associated estimates of uncertainty (e.g. confidence intervals) |
| ☐ | ☒ | For null hypothesis testing, the test statistic (e.g. $F$, $t$, $r$) with confidence intervals, effect sizes, degrees of freedom and $P$ value noted<br>*Give P values as exact values whenever suitable.* |
| ☒ | ☐ | For Bayesian analysis, information on the choice of priors and Markov chain Monte Carlo settings |
| ☒ | ☐ | For hierarchical and complex designs, identification of the appropriate level for tests and full reporting of outcomes |
| ☒ | ☐ | Estimates of effect sizes (e.g. Cohen's $d$, Pearson's $r$), indicating how they were calculated |

*Our web collection on statistics for biologists contains articles on many of the points above.*

## Software and code

Policy information about availability of computer code

| | |
|---|---|
| Data collection | Metabolomics: Masshunter Workstation Unknown Analysis software version 10.0 (Agilent) and the NIST 2017 MS library, MassHunter Workstation Quantitative Analysis software version 10.0 (Agilent).<br>NanoSIMS acquisition : Cameca Software NanoSIMS version 4.5 |
| Data analysis | Metabolomics: R studio (4.0.3) including the lmm2met v1.0 package;<br>Phylogenetic tree : MUSCLE v3.8.1551 by Robert C. Edgar; IQ-TREE multicore version 2.0.3 for Linux 64-bit built Dec 20 2020 and the ITOL online viewer (https://itol.embl.de/)<br>NanoSIMS: "L'image" software version 10-15-2021 (Larry Nittler, Carnegie Institution of Washington)<br>Codes used to analyze the data and generate figures are available at:<br>https://doi.org/10.5281/zenodo.10066636 |

For manuscripts utilizing custom algorithms or software that are central to the research but not yet described in published literature, software must be made available to editors and reviewers. We strongly encourage code deposition in a community repository (e.g. GitHub). See the Nature Portfolio guidelines for submitting code & software for further information.

# Data

Policy information about availability of data

All manuscripts must include a data availability statement. This statement should provide the following information, where applicable:

- Accession codes, unique identifiers, or web links for publicly available datasets
- A description of any restrictions on data availability
- For clinical datasets or third party data, please ensure that the statement adheres to our policy

> The kynureninase protein sequence was retrieved from the publicly accessible NCBI database (WP 025330329.1). Raw data is available on zenodo at the following link: https://doi.org/10.5281/zenodo.10066636

# Human research participants

Policy information about studies involving human research participants and Sex and Gender in Research.

| | |
|---|---|
| Reporting on sex and gender | N/A |
| Population characteristics | N/A |
| Recruitment | N/A |
| Ethics oversight | N/A |

Note that full information on the approval of the study protocol must also be provided in the manuscript.

# Field-specific reporting

Please select the one below that is the best fit for your research. If you are not sure, read the appropriate sections before making your selection.

☒ Life sciences    ☐ Behavioural & social sciences    ☐ Ecological, evolutionary & environmental sciences

For a reference copy of the document with all sections, see nature.com/documents/nr-reporting-summary-flat.pdf

# Life sciences study design

All studies must disclose on these points even when the disclosure is negative.

| | |
|---|---|
| Sample size | Sample sizes differ throughout as detailed in the Material and Methods section. Sample sizes were determined based on availability/opportunity, and are provided in each case as an exact sample size (n). |
| Data exclusions | Bacterial abundance quantification with qPCR: Samples with SD higher than 0.7 were excluded from the analysis<br>Metabolomics: Samples were removed from the metabolomics dataset with internal standards > or < 2*SD from the median of that batch. [Figure 1, n=6; Figure 5, n=3]. |
| Replication | Colonization experiments were successfully replicated five time using bees from separate hives as described in the methods. In vitro experiments were replicated independently 3-4 times. Metabolomic analysis of MF bee guts and 13C enrichment analysis of MF bees fed 13C glucose were not replicated. Feeding of MF bees with 13C glucose, followed by colonization with S. alvi was not replicated, but a prior pilot experiment to establish the correct experimental timing showed similar results. |
| Randomization | For every bee experiment, bees were randomly assigned to cages.<br>Metabolomics and qPCR samples were randomly extracted and analyzed after the experiments were conducted |
| Blinding | No blinding was performed for in vivo experiments, as the dietary condition (+/-Pollen) could not be masked. In vitro conditions were not blinded, but handled simultaneously in batches. |

# Reporting for specific materials, systems and methods

We require information from authors about some types of materials, experimental systems and methods used in many studies. Here, indicate whether each material, system or method listed is relevant to your study. If you are not sure if a list item applies to your research, read the appropriate section before selecting a response.

## Materials & experimental systems

| n/a | Involved in the study |
|-----|----------------------|
| ☒ | Antibodies |
| ☒ | Eukaryotic cell lines |
| ☒ | Palaeontology and archaeology |
| ☐ | ☒ Animals and other organisms |
| ☒ | Clinical data |
| ☒ | Dual use research of concern |

## Methods

| n/a | Involved in the study |
|-----|----------------------|
| ☒ | ChIP-seq |
| ☒ | Flow cytometry |
| ☒ | MRI-based neuroimaging |

# Animals and other research organisms

Policy information about studies involving animals; ARRIVE guidelines recommended for reporting animal research, and Sex and Gender in Research

| | |
|---|---|
| Laboratory animals | No laboratory animals were used in this study. |
| Wild animals | No wild animals were used in this study. |
| Reporting on sex | Newly emerged female workers bees were used. Results shown in the manuscript therefore apply only for female worker bees of the species Apis mellifera carnica. |
| Field-collected samples | Pupae of Apis mellifera carnica were removed from managed hives at the University of Lausanne. Pupae and emerged adult bees were reared in cages in temperature (33 C) and humidity (70% Rel.) controlled incubators and provided sucrose solution ad libitum. Prior to dissection, bees were anesthetized with CO2 and placed on ice. Wild bumble bees of species Bombus terrestris were caught on the campus of the university of Lausanne, sacrificed in the same manner as A. mellifera, and used to isolate bacterial strains. |
| Ethics oversight | No ethical approval was required for this study. |

Note that full information on the approval of the study protocol must also be provided in the manuscript.

