## [Peer Review File · Nature Microbiology]

Peer Review Information

Journal: Nature Microbiology

Manuscript Title: Host-derived organic acids enable gut colonization of the honey bee symbiont *Snodgrassella alvi*

Corresponding author name(s): Professor Philipp Engel

Reviewer Comments & Decisions:

Decision Letter, initial version:

Message: 3rd April 2023

Dear Philipp,

Thank you for your patience while your manuscript "Foraging on host synthesized metabolites enables the bacterial symbiont *Snodgrassella alvi* to colonize the honey bee gut" was under peer-review at Nature Microbiology. It has now been seen by 3 referees, whose expertise and comments you will find at the end of this email. Although they find your work of some potential interest, they have raised a number of concerns that will need to be addressed before we can consider publication of the work in Nature Microbiology.

In particular, referees #2 and #3 have relatively minor comments that should be straightforward to address. Referee #1 has an important concern regarding your use of microbiota depleted rather than germ free bees. We will require additional work to confirm that the microbiota depleted bees did not have low abundance bacteria that could influence your results, or additional experiments using germ free bees. We will not send this back to referees unless these data support the current results.

Should further experimental data allow you to address these criticisms, we would be happy to look at a revised manuscript.

Please include a data availability statement as a separate section after Methods but before

2references, under the heading "Data Availability". This section should inform readers about the availability of the data used to support the conclusions of your study. This information includes accession codes to public repositories (data banks for protein, DNA or RNA sequences, microarray, proteomics data etc...), references to source data published alongside the paper, unique identifiers such as URLs to data repository entries, or data set DOIs, and any other statement about data availability. At a minimum, you should include the following statement: "The data that support the findings of this study are available from the corresponding author upon request", mentioning any restrictions on availability. If DOIs are provided, we also strongly encourage including these in the Reference list (authors, title, publisher (repository name), identifier, year). For more guidance on how to write this section please see:

<http://www.nature.com/authors/policies/data/data-availability-statements-data-citations.pdf>

* If you have not done so already we suggest that you begin to revise your manuscript so that it conforms to our Article format instructions at <http://www.nature.com/nmicrobiol/info/final-submission>. Refer also to any guidelines provided in this letter.

When submitting the revised version of your manuscript, please pay close attention to our [href="https://www.nature.com/nature-portfolio/editorial-policies/image-integrity">Digital Image Integrity Guidelines. and to the following points below:](https://www.nature.com/nature-portfolio/editorial-policies/image-integrity)

Note: This url links to your confidential homepage and associated information about manuscripts you may have submitted or be reviewing for us. If you wish to forward this e-mail to co-authors, please delete this link to your homepage first.

Nature Microbiology is committed to improving transparency in authorship. As part of our efforts in this direction, we are now requesting that all authors identified as 'corresponding author' on published papers create and link their Open Researcher and Contributor Identifier (ORCID) with their account on the Manuscript Tracking System (MTS), prior to acceptance. This applies to primary research papers only. ORCID helps the scientific community achieve unambiguous attribution of all scholarly contributions. You can create and link your ORCID from the home page of the MTS by clicking on 'Modify my Springer Nature account'. For more information please visit www.springernature.com/orcid.

If you wish to submit a suitably revised manuscript we would hope to receive it within 6 months. If you cannot send it within this time, please let us know. We will be happy to consider your revision, even if a similar study has been accepted for publication at Nature Microbiology or published elsewhere (up to a maximum of 6 months).

Yours sincerely,

[redacted]

Reviewer Expertise:

Referee #1: bee microbiome, symbiosis
Referee #2: metabolomics, microbial metabolism, symbiosis
Referee #3: insect nutritional symbioses

Reviewer Comments:

Reviewer #1 (Remarks to the Author):

Summary

The authors present an intriguing story about how *Snodgrassella alvi*, a honey bee gut symbiont incapable of glycolysis, colonizes the gut when the rest of the microbiota is depleted. Past studies have suggested a potential cross-feeding relationship between *Gilliamella* and *Snodgrassella*. The authors asked the questions (1) Does co-colonization with *Gilliamella* improve *Snodgrassella* colonization? (2) How does *Snodgrassella*-supplementation change the metabolome? And (3) what is *Snodgrassella* using as a carbon source and is it host or microbiome-derived? Interestingly, the addition of *Gilliamella* to newly-eclosed bees supplemented with *Snodgrassella* did not increase *Snodgrassella*'s titer. However, in this experiment (as well as the metabolomics of supplemented bees and NanoSIMS) microbiome-depleted bees

3are conflated with germ-free bees. This experimental design muddies some of the author's findings and major revisions are necessary to reflect this in the text. The questions asked in this narrative are deserving of further exploration and the methodology used is appropriate (besides the use of MD bees) and well-thought out. With more careful language and framing this will be a more compelling story.

Major Revisions:

Host-derived vs microbiome-derived:

The premise of most experiments in this manuscript is that metabolites found in the microbiome-depleted (MD) bees are the result of host metabolism alone. However, it cannot be ruled out that low abundance members are contributing to cross-feeding with *Snodgrassella*. Many other studies show that newly-eclosed bees have not acquired most of the core adult gut microbiome, but that they are colonized by other microbes. Sequencing of the microbiome of MD bees is not included in this study but qPCR with *Snodgrassella* and *Gilliamella* specific primers shows at least 10^4 16S rRNA copies on average per bee. The CFU data in the supplement supports this, showing at least 10^2 cells on average. Although this is certainly depleted compared to conventionally-reared bees, these microbes could certainly be contributing to *Snodgrassella*'s growth.

To address this concern the authors either need to show (1) that the MD bees are germ-free or (2) reframe the narrative so it is clear that nutrition could be microbiome-derived.

Misleading figures and terminology:

There are several figures in which the axes have either misleading labels or scales (listed below).

- Figure 1A and S1B: "# bacteria/gut" is used when it is actually # of *Snodgrassella* (or *Gilliamella*) genome number equivalents. This labelling infers there are no bacteria present based on a non-targeted approach.
- Figure 1A, Figure 5A, Figure S1 A & B: The y-axes start at 10^2 instead of 0, so it appears that there are no detectable CFUs or genome copies in the MD samples. Changing the axes to begin at 0 would better reflect the findings.
- "Limit of detection": on line 597 the authors state that 500 cells is the limit of detection for CFU counting. More explanation is needed here since fewer than 500 bacterial cells can normally be counted by plating.

Introduction of pollen-associated microbes:

In experiments where pollen was fed to newly-emerged bees, it was not sterilized. Pollen and nectar have microbial associations of their own in which pollen can be broken down, releasing nutrients. With this in mind, it is unclear whether differentially-produced metabolites between sugar and pollen-fed groups is due to differences in bee's metabolism or if it is due to the addition of new microbial members.

To address this the authors should either (1) repeat the experiments with sterilized pollen or (2) acknowledge that pollen-associated microbes could contribute to the metabolome.

NanoSIMS of *Snodgrassella*

Given the concerns regarding MD bees, it would be nice to either (1) see images of an

earlier time point in the NanoSIMS pulse-chase experiment to confirm there are no bacterial cells present before *Snodgrassella* supplementation or (2) sequence guts to confirm that only *Snodgrassella* is present.

Minor revisions:

-Link to Bee9 media is broken.

-In Fig. 2B was there any enrichment in acetate? Since the authors later hypothesize in the discussion that acetate could be used by *Snodgrassella* as a carbon source, it would be nice to see that reflected here.

Reviewer #2 (Remarks to the Author):

Review for Nature Microbiology MS NMICROBIOL-23010206 " Foraging on host synthesized metabolites enables the bacterial symbiont *Snodgrassella alvi* to colonize the honey bee gut" by Quinn et al.

The Ms. by Quinn et al. reports a detailed metabolism study of the symbiosis between bees as host and their specific gut symbionts using untargeted metabolomics, stable isotope metabolomics and high resolution cell activity measurements using nanoSIMS.

The main finding suggests one bacterial strain from the gut, *Snodgrassella* is adapted to a specific metabolic niche in the gut. The scientific novelty is arguably the fact that the gut symbiont does not live off the main resource from the bee diet, namely sugars.

Snodgrassella depends on host-derived resources which are mainly organic acids produced by the host from the provided sugars. This observation could be a general pattern for metabolic interactions and in my opinion symbiosis researchers across many different models should have this extra loop of nutritional/metabolic interaction in mind for future studies. The importance of host secreted metabolites which benefit the microbial community is so far not well studied and Quinn et al. provide very good evidence using elegant experimental approaches and point towards certain metabolite classes to look for.

The authors give a brief and direct introduction into the topic of bee gut symbioses and made it easy to dive into their experiments and results.

Lots of results from a combination of in-vivo and growth experiments. Most times very well explained, some figures and results need more clarity (e.g. Fig1C would benefit from 1 or 2 metabolites shown as box plots or similar to see the quant. Changes across conditions.

General comments and questions:

Does *S. alvi* growth not at all on sugars? See further down comments on fig4

Why was ^{13}C glucose used for the pulse chase and not ^{13}C Sucrose? Sucrose was the carbon source for all previous experiments. Did the authors have evidence that the fructose from hydrolyzing sucrose has no effect? I think it should be mentioned for transparency that it could induce different phenotypes.

5The in vitro assays are a clear benefit for the study. Did the authors perform any experiments with mixed substrates? Does *S. alvi* use sugars in the presence of organic acids? Figure S4 indicates this?

I think the anthranilate, kynurenic acid and tryptophan examples are very nice confirmations of in-vivo observations. Although it is interesting to learn that *S. alvi* is also excreting similar organic acids in the media, which in turn are predicted to come from the host and feed the symbiont. Do the authors have results which could account for the ratio of host related vs symbiont related organic acids in-vivo?

The description of the Bee9 medium is missing some details- Are there amino acids present during the incubation experiments with the different organic acids? Why was the citric acid component not exchanged with the 'sole carbon source' aka the organic acids tested for growth- I think this is slightly misleading.

I tried to understand figure 4C with the blue and orange, but failed at first. The 'products' indicate production, but the blue squares are compounds added. Why not leave them out? Formate, Butyrate and Propanoate have no filled square at all- should this indicate 'no production'?

Interesting to see that condition of 'all' substrates added actually is lower compared to just HMG- would you like to speculate which metabolite inhibits? Was the medium composition adjusted to C -input (molar)? Adding 2C like acetate makes a difference compared to a C6 compound like HMG.

HMG seems central and its not a commonly known organic acid- could the authors give some background please?

I think in Fig4B a mix-up happened for the citric acid label on glucose and hypoxanthin- please check? There is one condition citric acid - glucose which shows growth, needs clarification as this is an important combination comparing the conditions (in-vivo)

Figure 3

The nanoSIMS results are very intriguing and show the real in-vivo activity of symbiont cells. How do the authors explain that those cells are *S.alvi* – based on microbial sequences of the gut content? Or did they confirm via FISH or other correlative imaging methods? I think the measurements clearly show incorporation of ^{13}C , but are these *S. alvi* cells? Is there a way to combine the qPCR results with the nanoSIMS data- I expected more cells in the gut (TEM images) based on the cell numbers reported in fig1A. But maybe the authors have evidence from other studies if this is the 'normal' distribution/cell number.

Please indicate in legend EAA and NEAA – although very cool to see the essential amino acids unlabelled and other metabolites labeled at approx. 50%. Although the percentage of organic acids surprises (I expected much high labelling after the glucose pulse).

In dataset1 the % labeling is very different within the group of NEAA, I would not summarize them all as one group average, rather show exemplary (maybe a high and low

example)

The supplement figure S4 indicates high consumption of certain carbon sources, also caffeate. This compound is listed also in fig4B and its not clear if growth happened or not, also not obvious why its indicated as carbon source.

Discussion is very well balanced and carefully phrased statements including naming of limitations. The discussion around L287: nutrients of pollen have minor impact compared to host. Which data supports this claim? Is there nanoSIMS data on *S. alvi* after the pulse of ¹³C glucose plus unlabelled pollen?

Small points:

L45 - These phylotypes are culturable – all of the 8-10?

L50 – starches – I think starch is one sugar polymer and does not summarize different types

L54 – biomass through gluconeogenesis. provide ref please

L68 – these compounds – I think you can be already more specific here

L106 – and an unknown – specify the unknown here (unknown_RT)

L136 – synthesized in adult bees – should be a statement like: not synthesized during the course of experiment

L140 – I like the careful statement and explanation of possible source of isotope dilution

Caffeic acid or caffeate, authors should check the Ms. for the nomenclature of all metabolites and try to be consistent. In a few places its off.

The authors did very well on reporting the replication and provide sufficient datasets. The GitHub link to the raw data did not work for me and I would like to see the metabolomics data deposited on metabolights or similar database.

Reviewer #3 (Remarks to the Author):

Review of Quinn et al: Foraging on host synthesized metabolites enables the bacterial symbiont *Snodgrassella alvi* to colonize the honey bee gut

This paper by Quinn et al., seeks to understand whether host-produced metabolites feed *Snodgrassella*, a gut symbiont of honey bees. The authors tackle their question using a series of approaches that include manipulative diet experiments coupled with metabolome analysis, stable isotope labelling experiments, NanoSIMS, in vitro growth assays, examination of genetic variation in the colonization success of *Snodgrassella* in honey bee guts under a range of dietary conditions. Taken together the data show that *Snodgrassella* feeds on host synthesized metabolites, a result that has exciting implications for understanding the establishment and evolution of this symbiosis.

7I have two main concerns about the work.

1/ qPCR quantification of *Snodgrassella* and other symbionts population sizes rests on the assumption that the bacteria are not polyploid. It is common for symbiont population size and symbiont genome copy number to be uncoupled because the ploidy of symbionts is not fixed or constant see:

Komaki, K., and Ishikawa, H. (2000). Genomic copy number of intracellular bacterial symbionts of aphids varies in response to developmental stage and morph of their host. *Insect. Biochem. Mol. Biol.* 30, 253–258. doi: 10.1016/s0965-1748(99)00125-3

Mergaert, P., Uchiumi, T., Alunni, B., Evanno, G., Cheron, A., Catrice, O., et al. (2006). Eukaryotic control on bacterial cell cycle and differentiation in the *Rhizobium*-legume symbiosis. *Proc. Natl. Acad. Sci. U.S.A.* 103, 5230–5235. doi: 10.1073/pnas.0600912103

Woyke, T., Tighe, D., Mavromatis, K., Clum, A., Copeland, A., Schackwitz, W., et al. (2010). One bacterial cell, one complete genome. *PLoS One* 5:e10314. doi: 10.1371/journal.pone.0010314

The variable ploidy of symbionts means that it is impossible to obtain meaningful estimates of symbiont population sizes using qPCR unless the authors demonstrate that *Snodgrassella* are haploid.

Given that the authors also use culture-based methods of quantifying *Snodgrassella* populations, I suggest dropping the qPCR estimates from the manuscript. The overall results will not be changed by dropping the qPCR data.

2/Presentation of the metabolome data and analyses need to be improved. To make it more accessible to people unfamiliar with these types of analyses. I found the work presented in Figure 1C particularly hard to understand with respect to the directionality of the relationships/contributions. I am not convinced that these data are needed to address the central question of the paper as it is currently presented.

The most compelling results are the labelling experiments.

Detailed comments on the manuscript to assist in improving the work:

- Line 73 – it would be good to give a brief description of how microbiota depleted honey bees were generated. I also suggest avoiding use of the abbreviation MD – and just refer to them as microbiota depleted through the manuscript
- Line 74 -75 : reword “alone or with both” this is confusing
- Line 75 and throughout – pollen, from what plant? I assume that this matters and that the metabolic profile of pollen from different plants varies. If not, state that all pollen has the same metabolic profile.
- Line 135: it would be good in here and likely other places to be clear when something is not expected, for example, the result that essential amino acids are not enriched is entirely unsurprising because bees can’t make essential amino acids. I suggest looking carefully through the whole manuscript to be clearer about what the null expectations for each experiment are as this will help the reader understand better when a result is consistent with expectations and when it is not.
- Line 321 – what is the source of the pollen?

- Line 333 – frozen at what temperature?
- Line 337 – explain that U-13C is universally labelled, what does that mean?
- Line 341 – was the 12C glucose also universally labelled?
- Line 358 – replace “an” with “a”
- Line 363 – italics for species name
- Line 367 and many other places – use the micro symbol not a letter u
- Line 390 and other places – mL not ml
- Line 463 – replace , with . in number
- Figure 1A – significance missing for 3rd data set across
- Figure 1C – needs improved labelling and explanation to increase the accessibility of these data to the reader. E.g. for cross-feeding include an indication of the direction in the figure. Better label the figure so you don’t have to read the legend to find out what the red vs. blue dots mean. The dotted lines are light, maybe shade the area within the lines? Line 517 – lower-left section – maybe also shade this area??
- Fig 3B – line is too faint
- Fig 3E - crop is invisible
- Fig 4B – add a key to the figure on the figure
- Line 562 – did you perform a correction for multiple tests??
- Fig 5A – add a bar across the top marking those that are native to Apis and those that are native to Bombus.
- Fig 5B – the colors of 1 and 5 are too similar
- Fig 6 – blue arrows are substrates? Aren’t they enzymes? Substrates in boxes??
- Line 588 – in what genome?
- Fig S2 – text in panels is too small to read
- Fig S4 – for glucose at 0% the dots are aligned on the horizontal to indicate that three data points were taken. Is this also true for those substrates that reach 100% or do these represent just a single data point?
- Figure S5 – names of taxa too small to read

Author Rebuttal to Initial comments

Reviewer Comments:

Reviewer #1 (Remarks to the Author):

Summary

The authors present an intriguing story about how *Snodgrassella alvi*, a honey bee gut symbiont incapable of glycolysis, colonizes the gut when the rest of the microbiota is depleted. Past studies have suggested a potential cross-feeding relationship between *Gilliamella* and *Snodgrassella*. The authors asked the questions (1) Does co-colonization with *Gilliamella* improve *Snodgrassella* colonization? (2) How does *Snodgrassella*-supplementation change the metabolome? And (3) what

9is Snodgrassella using as a carbon source and is it host or microbiome-derived? Interestingly, the addition of Gilliamella to newly-eclosed bees supplemented with Snodgrassella did not increase Snodgrassella's titer. However, in this experiment (as well as the metabolomics of supplemented bees and NanoSIMS) microbiome-depleted bees are conflated with germ-free bees. This experimental design muddies some of the author's findings and major revisions are necessary to reflect this in the text. The questions asked in this narrative are deserving of further exploration and the methodology used is appropriate (besides the use of MD bees) and well-thought out. With more careful language and framing this will be a more compelling story.

Major Revisions:

Host-derived vs microbiome-derived:

The premise of most experiments in this manuscript is that metabolites found in the microbiome-depleted (MD) bees are the result of host metabolism alone. However, it cannot be ruled out that low abundance members are contributing to cross-feeding with Snodgrassella. Many other studies show that newly-eclosed bees have not acquired most of the core adult gut microbiome, but that they are colonized by other microbes. Sequencing of the microbiome of MD bees is not included in this study but qPCR with Snodgrassella and Gilliamella specific primers shows at least 10^4 16S rRNA copies on average per bee. The CFU data in the supplement supports this, showing at least 10^2 cells on average. Although this is certainly depleted compared to conventionally-reared bees, these microbes could certainly be contributing to Snodgrassella's growth.

To address this concern the authors either need to show (1) that the MD bees are germ-free or (2) reframe the narrative so it is clear that nutrition could be microbiome-derived.

Reply: Thank you for the concise summary of our aims and results. We agree that the question of sterility is central to the analysis and interpretation of the paper and hence we fully understand the concern of this reviewer. In the following, we provide supportive explanations and new experimental data to show that our bees did not contain relevant levels of viable microbes.

1. We wish to apologize for the confusion generated by our use of the term "Microbiota-depleted" to describe bees in this experiment. We chose this term to differentiate the

10experimental differences in honey bee research methods from those of germ-free or microbiota-free rodent models. However, we concede that it creates confusion for readers. To be clear, our method exactly follows established protocols in the field of bee microbiota research for generating what has been described as “microbiota-free” or “germ-free” bees [e.g. Leonard et al Science; Zheng et al, PNAS; Kwong et al, PNAS]. Importantly, we did not work with newly-eclosed bees that emerge on the brood frame and could indeed become contaminated by ingesting environmental bacteria [Powell et al Appl Environ Microbiol; Martinson et al Appl Environ Microbiol]. As stated in the methods section “Experimental colonization of gnotobiotic bees”, we instead aseptically removed pupae to emergence boxes in lab incubators at an early developmental stage before they fully developed their digestive system. As bee pupae shed their larval gut intema, their developed digestive tract is usually microbiota-free until they begin eating. Moreover, we forgot to mention in the methods that all diets used in the experiments were sterile-filtered or sterilized by irradiation. To avoid any misunderstanding and to make our manuscript more consistent with the nomenclature used in previous studies we replaced the term microbiota-depleted (MD) by microbiota-free (MF) throughout the manuscript.

2. The reviewer is right that despite all precaution, microbiota-free bees raised from extracted pupae in the laboratory can, on occasion, be contaminated with low amounts of bacteria or yeast [see e.g. Tauber et al, Yeast; Liberti et al, Nat Ecol Evol; Martinson et al Appl Environ Microbiol]. While this concerns the minority of newly emerged bees reared in the way explained above, we usually try to identify and exclude such bees from experiments. We have done this for all experiments presented in the manuscript by plating guts from 2 newly emerged bees from each emergence box. We only continued with that experimental replicate if the plates were clean. All newly emerged bees were then housed in sterilized cages, handled with sterile tools, and fed sterilized food. At the end of the experiment, we again checked for contamination in sampled bees. We wish to stress that we detected no live cells via CFU plating (Figure S1A, Figure 5A) in the microbiota-free bees, and we detected no contaminating bacteria in our colonized bees. The latter case would manifest in bacterial growth on plates other than BHIA. We have included these details in the Method section of the revised manuscript on Line 360.

3. The TEM images of ileum sections show no signs of microbial contamination. At times 24h and 32h after inoculation we measured low *S. alvi* cell numbers via CFU plating (Median

< 10^3 CFU/Ileum). Careful examination of the TEM images revealed no visible bacterial cells in the ileum at those timepoints. See Dataset S2.

4. To provide additional evidence from culture-independent analysis, we have now carried out qPCR with a universal bacteria (16S rRNA gene amplicon) and a fungal (18S rRNA gene amplicon) primer set on the bee guts of the original colonization experiment (experiment shown in Figure 1). Unfortunately, pollen grains confound the analysis, as pure pollen amplifies to high levels for both primer sets, and bees that consumed pollen averaged $\sim 10^6$ 16S rRNA gene copies per gut. This has been noted already in previous studies and can be explained by chloroplast amplification. However, in bees fed sugar water only, the 16S PCR assays showed only four bees with detectable signal (above the technical limits of detection) in the MF condition, and copy numbers in three of the four were ~ 1000 fold less than in colonized bees. The fungal primer set resulted in consistent signals in almost all our bee samples, but the values were only marginally above the limit of detection and no live fungi could be recovered from any of our bees (see next point). We have added this data to the revised manuscript in the text on Line 84 and Figure S1. Pure pollen grains also amplified the 18S amplicon to high levels, and we detected persistent, low amplification (CT values ~ 30) across all bees.

5. We carried out an additional experiment during the revision, to provide further evidence that newly emerged bees reared as described above are indeed microbiota-free. We generated microbiota-free bees as before and plated their gut content at day zero and six post emergence on a wide range of different media that were incubated in different conditions. This included media specifically tailored towards fungal and yeast cultivation. As in our previous experiments no bacterial/fungal/yeast growth was observed at either timepoint. Line 132.

6. From the same experiments, we also determined the absolute organic acid concentrations in the gut. We saw that despite the absence of microbial contamination, the levels of many key metabolites utilized by *S. alvi* increased between day zero and day six (Figure 2A). We also wish to highlight that combining the results shown in Figures 2A, 2B, 3D, and S5 presents a strong biochemical case for the bee as the source of these metabolites. The metabolomic results show an increase in the abundance of essential amino acids including tryptophan, phenylalanine, and methionine, as well as of kynurenine, a product of tryptophan metabolism. However, unlike other amino acids and organic acids, these

12compounds do not become ^{13}C labelled after feeding ^{13}C glucose. We expect this in the bee, as it cannot *de novo* synthesize essential amino acids from glucose. Instead, it recycles them from its own biomass, which was previously assimilated from the diet during larval development. The same phenomenon would not be true in yeast or bacteria that maintain the ability to synthesize most amino acids. In such a case, all amino acids and their downstream products would become substantially ^{13}C enriched. (Note: This experiment has replaced our previous analysis in the original Figure 2A, as the results directly address the question as to the metabolic origin of metabolites in the diet vs from the host)

7. Our experimental results encompass, in total, 12 independent replicates in which we colonized microbiota free bees from different hives at different times. We show that the host associated organic acids that *S. alvi* utilizes were detected across all samples. Such results are not consistent with an alternate hypothesis that transient and variable environmental contaminants synthesized these compounds. In such a case we would expect to find much more variation in the organic acid profiles across bees.

8. The reviewer noted that we plotted 'negative' qPCR values just below 10^4 genome copies and CFU values just below 10^2 CFUs. This does not mean that we detected 10^4 or 10^2 bacteria in our experiments, but it indicates our limit of detection, i.e. we do not have the resolution to detect anything which is smaller than that threshold. For example, we did not plate the entire bee gut for CFU enumeration, but only 10 μl of 700 μl gut homogenate. This means our limit of detection is 1 CFU in 10 μl . As we only plated 1/70 of an entire gut, our limit of detection for the entire gut is thus 1 CFU \times 70 = 70 CFUs. The rationale is applied to the qPCR data. We usually only use 1 μl of a 50 μl reaction. Water usually amplifies at 10 copies. So, our limit of detection is 10 copies \times 50 = 500 copies.

We hope that the additional experimental data (qPCR data with universal primers, additional experiment in which we plated on a wide range of media and determined organic acid concentrations) together with our additional clarifications alleviate remaining concerns of the reviewer that the source of the organic acids is not the host.

Misleading figures and terminology:

There are several figures in which the axes have either misleading labels or scales (listed

13below).

-Figure 1A and S1B: “# bacteria/gut” is used when it is actually # of *Snodgrassella* (or *Gilliamella*) genome number equivalents. This labelling infers there are no bacteria present based on a non-targeted approach.

Reply: The Figure has been updated to improve clarity for the reader.

-Figure 1A, Figure 5A, Figure S1 A & B: The y-axes start at 10^2 instead of 0, so it appears that there are no detectable CFUs or genome copies in the MD samples. Changing the axes to begin at 0 would better reflect the findings.

Reply: It is in fact true that we detected no bacterial CFU's in all microbiota free samples. The values are plotted just below the experimental limits of detection to reflect the true measurement ability. After discussion, we decided that we could most accurately and clearly represent the data by plotting the LOD as the minimal value on the y-axis. This means that no numerical value is shown for datapoints below that threshold. Figure 1, Figure 3, Figure 5, and Figure S1 have been updated accordingly.

-“Limit of detection”: on line 597 the authors state that 500 cells is the limit of detection for CFU counting. More explanation is needed here since fewer than 500 bacterial cells can normally be counted by plating.

Reply: We thank the reviewer for highlighting this typo. We had mentioned the limit of detection of the qPCR assay instead of the CFU plating. The true LOD was 70 CFU/gut as we plated 10 μ L of gut homogenate initially suspended in 700 μ L PBS solution. We could not plate the entire gut content on a single plate, as we needed aliquots for multiple plates (MRSA, LBA, BHIA) as well as DNA extraction and metabolomics.

Introduction of pollen-associated microbes:

In experiments where pollen was fed to newly-emerged bees, it was not sterilized. Pollen and nectar have microbial associations of their own in which pollen can be broken down, releasing nutrients. With this in mind, it is unclear whether differentially-produced

14metabolites between sugar and pollen-fed groups is due to differences in bee's metabolism or if it is due to the addition of new microbial members.

To address this the authors should either (1) repeat the experiments with sterilized pollen or (2) acknowledge that pollen-associated microbes could contribute to the metabolome.

Reply: We regrettably neglected to specify that pollen used in these experiments was sterilized via electron beam irradiation. The methods have been updated to reflect this (Line 343). Following sterilization, the pollen samples tested negative for live contamination after incubating pollen granules in PBS solution and plating them on LBA, NA, BHIA, MRSA, and CBA+5%SB media and incubating the plates in aerobic, microaerophilic, and anaerobic conditions.

NanoSIMS of *Snodgrassella*

Given the concerns regarding MD bees, it would be nice to either (1) see images of an earlier time point in the NanoSIMS pulse-chase experiment to confirm there are no bacterial cells present before *Snodgrassella* supplementation or (2) sequence guts to confirm that only *Snodgrassella* is present.

Reply: We have included more TEM images of ileum sections in Dataset S2. In these images one can see that no bacterial cells are visible at early timepoints in both colonized and microbiota-free control bees. Only for late timepoints (i.e. 48 h and 72 h post colonization) and only in case of bees colonized with *Snodgrassella* we could observe microbial cells in the TEM images. Also, with the additional experimental data and more careful explanations of our methodology (as described in our eight-point reply to the major concern of this reviewer), we believe we have provided sufficient lines of evidence that the microbiota-free bees do not harbor any relevant levels of live microbes.

Minor revisions:

-Link to Bee9 media is broken.

Reply: Thank you. The link has been fixed.

-In Fig. 2B was there any enrichment in acetate? Since the authors later hypothesize in the

discussion that acetate could be used by *Snodgrassella* as a carbon source, it would be nice to see that reflected here.

Reply: We only analyzed the ^{13}C labelling using the soluble metabolite extraction method in this experiment. This precluded analyzing acetate, as it was only detectable with the method for measuring short chain fatty acids. We made this choice, because acetate was barely detectable in guts of MF bees fed sugar water alone. Our data from the first colonization experiment indicated that it instead comes from *Gilliamella* and pollen, while *S. alvi* is also able to produce it under certain conditions, namely with 3Hmg or pyruvate as a major substrate.

Reviewer #2 (Remarks to the Author):

Review for Nature Microbiology MS NMICROBIOL-23010206 " Foraging on host synthesized metabolites enables the bacterial symbiont *Snodgrassella alvi* to colonize the honey bee gut" by Quinn et al.

The Ms. by Quinn et al. reports a detailed metabolism study of the symbiosis between bees as host and their specific gut symbionts using untargeted metabolomics, stable isotope metabolomics and high resolution cell activity measurements using nanoSIMS.

The main finding suggests one bacterial strain from the gut, *Snodgrassella* is adapted to a specific metabolic niche in the gut. The scientific novelty is arguably the fact that the gut symbiont does not live off the main resource from the bee diet, namely sugars.

Snodgrassella depends on host-derived resources which are mainly organic acids produced by the host from the provided sugars. This observation could be a general pattern for metabolic interactions and in my opinion symbiosis researchers across many different models should have this extra loop of nutritional/metabolic interaction in mind for future studies. The importance of host secreted metabolites which benefit the microbial community is so far not well studied and Quinn et al. provide very good evidence using elegant experimental approaches and point towards certain metabolite classes to look for.

Reply: Thank you for the kind words regarding our study. We agree that host derived compounds may play a far greater role than currently appreciated in sustaining their native gut microbiota.

The authors give a brief and direct introduction into the topic of bee gut symbioses and made it easy to dive into their experiments and results.

Lots of results from a combination of in-vivo and growth experiments. Most times very well explained, some figures and results need more clarity (e.g. Fig1C would benefit from 1 or 2 metabolites shown as box plots or similar to see the quant. Changes across conditions.

Reply: Box plots of normalized relative concentrations (z-scores) are shown for metabolites significantly affected by *S. alvi* colonization in Figure S3.

General comments and questions:

Does *S. alvi* growth not at all on sugars? See further down comments on fig4

Reply: No, *Snodgrassella* cannot grow on sugars as it lacks a complete pathway for utilizing them. The key missing genes include (EMP glycolysis: Phosphofructokinase EC 2.7.1.11; Pentose phosphate pathway: glucose-6-phosphate 1-dehydrogenase, EC 1.1.1.49; Entner–Doudoroff pathway: Completely absent) [Kwong et al, PNAS]. As one can see from our *in vitro* growth experiment, when providing glucose, no growth is detected. We haven't tested other sugars in this paper, but based on the genome annotation as well as our unpublished data from Biolog PM1 carbon source growth plates, we have no evidence that growth on saccharides is possible.

Why was ¹³C glucose used for the pulse chase and not ¹³C Sucrose? Sucrose was the carbon source for all previous experiments. Did the authors have evidence that the fructose from hydrolyzing sucrose has no effect? I think it should be mentioned for transparency that it could induce different phenotypes.

Reply: Our sugar water experimental condition is already artificial compared to a normal bee diet, in the sense that the saccharide complexity and diversity is greatly reduced. While the bee physiology and metabolism most likely changes in response to feeding of different

17sugars, the purpose of the labelling experiments was to show that the bee can metabolize saccharides into organic acids that are released into the gut. For that reason, we only report their isotopic enrichment and do not compare metabolite abundances between experiments. In our pilot proof of concept experiments (data not reported) we utilized ^{13}C glucose because U- $^{13}\text{C}_{12}$ sucrose is 15 times more expensive, and we needed to feed a substantial quantity of sugar. Given the high degree of labelling in bee synthesized compounds we saw little need to switch to ^{13}C sucrose in our primary experiments.

The in vitro assays are a clear benefit for the study. Did the authors perform any experiments with mixed substrates? Does *S. alvi* use sugars in the presence of organic acids? Figure S4 indicates this

Reply: We first utilized single carbon source experiments, but found that *S. alvi* could not grow on all of the putative substrates. In those instances, we re-tested growth with the addition of citrate to the media (*i.e.* a dual substrate experiment). We also mixed all of the substrates together in equal proportions as a pseudo rich media that more closely resembles the environment in the gut. As shown in Figure 4B, the substrates that couldn't be utilized individually all boosted growth compared to condition with citrate alone in the media. We show in Figure 4B that glucose does not serve as a sole carbon source, and that it does not enhance growth, and then we further show in updated Figure S6 that glucose is not depleted from the medium by *S. alvi*.

I think the anthranilate, kynurenic acid and tryptophan examples are very nice confirmations of in-vivo observations. Although it is interesting to learn that *S. alvi* is also excreting similar organic acids in the media, which in turn are predicted to come from the host and feed the symbiont. Do the authors have results which could account for the ratio of host related vs symbiont related organic acids in-vivo?

Reply: Our results indicated to us that many, or indeed most, metabolites of interest have multiple origins and sinks. This is particularly true for metabolites closely related to central carbon and energy metabolism. Therefore, it is exceedingly difficult to quantitatively map fluxes or assign relative contributions to host, diet, and microbes. Acetate is a particularly good example of this phenomenon. We were puzzled at first when our colonization experiment clearly showed that *Gilliamella* produced acetate, and *S. alvi* consumed it under mono-colonization conditions with pollen present; but when co-colonized together acetate

18levels actually increased. Our *in vitro* experiments provided an explanation, as *S. alvi* facultatively produces acetate when consuming 3Hmg or pyruvate, while *Gilliamella* synthesizes pyruvate and does not compete for 3Hmg as a substrate. Such a condition-specific metabolic shift could possibly be measured with an elegant labelling experiment, but this creates a catch-22 paradox: the exotic nature of the experiment needed to capture *in vivo* fluxes would be so far removed from the normal physiological conditions that the quantitative information would not be reflective of the normal physiology.

The description of the Bee9 medium is missing some details- Are there amino acids present during the incubation experiments with the different organic acids? Why was the citric acid component not exchanged with the 'sole carbon source' aka the organic acids tested for growth- I think this is slightly misleading.

Reply: We have fixed the link to the complete Bee9 media recipe in the paper. Our explanation in the manuscript is accurate: "*We next tested whether the putative host synthesized substrates were sufficient for growth of S. alvi as sole carbon sources in a chemically defined liquid medium "Bee9"*". The only carbon in the medium comes from the specified organic acids that we add. We tested all organic acids as sole carbon sources, but in cases where they did not support growth, we then added them to Bee9 + citrate to see if they could enhance growth. Only three metabolites (citrate, isocitrate, and malate) supported growth as sole carbon sources in this condition. However, the addition of 3Hmg, fumarate, succinate, gamma-aminobutyrate (Gaba), kynurenine, or urea to Bee9 + citrate significantly improved the maximal growth rate of *S. alvi* versus with citrate alone.

I tried to understand figure 4C with the blue and orange, but failed at first. The 'products' indicate production, but the blue squares are compounds added. Why not leave them out? Formate, Butyrate and Propanoate have no filled square at all- should this indicate 'no production'?

Reply: We have updated Figure 4C in multiple ways. In particular, we now show the log₂(fold change) between the spent and fresh media. Formate, butyrate, and propanoate were indeed not detected in any samples. We included them to indicate that our analysis did look for their production, as they are important microbially synthesized SCFA's. However, we have now removed them from the plot for clarity.

Interesting to see that condition of 'all' substrates added actually is lower compared to just HMG- would you like to speculate which metabolite inhibits? Was the medium composition adjusted to C -input (molar)? Adding 2C like acetate makes a difference compared to a C6 compound like HMG.

Reply: Carbon sources were added in equimolar amounts for experimental ease, but as the reviewer noted, this does not equal equimolar carbon levels. For this reason, we use the maximum exponential growth rate, rather than final OD or area under the curve as our readout for growth, as those final measures are subject to carbon limitations in the growth media. It is possible that growth on all substrates is lower than with citrate + 3Hmg because acetate was one of the substrates added to the media, but is also the major byproduct of 3Hmg metabolism. This could lead to product inhibition of growth stimulating 3Hmg metabolism, but would require further experimental validation to state with any confidence.

HMG seems central and its not a commonly known organic acid- could the authors give some background please?

Reply: We agree that is a rather obscure molecule in this research context. In the Discussion we have provided some information about its function in known host metabolic pathways "3Hmg is an intermediate of the host's isoprenoid biosynthesis, leucine degradation and ketone body metabolism, though why this metabolite is released into the gut remains elusive." We are currently performing follow-up experiments to further uncover its importance for *S. alvi* though at the moment do not have more information to add.

I think in Fig4B a mix-up happened for the citric acid label on glucose and hypoxanthin- please check? There is one condition citric acid - glucose which shows growth, needs clarification as this is an important combination comparing the conditions (in-vivo)

Reply: We thank the reviewer for spotting this formatting mistake with the label "hypoxanthine", which resulted in a shift of the subsequent x-axis labels. The figure has been corrected.

Figure 3

The nanoSIMS results are very intriguing and show the real in-vivo activity of symbiont cells.

20How do the authors explain that those cells are *S. alvi* – based on microbial sequences of the gut content? Or did they confirmed via FISH or other correlative imaging methods? I think the measurements clearly show incorporation of ^{13}C , but are these *S. alvi* cells? Is there a way to combine the qPCR results with the nanosSIMS data- I expected more cells in the gut (TEM images) based on the cell numbers reported in fig1A. But maybe the authors have evidence from other studies if this is the 'normal' distribution/cell number.

Reply: We did not combine the NanoSIMS images with FISH tags as this was a monocolonization experiment of microbiota-free hosts. Here, we have confidence in the bacterial identity, as the cell shape and size observed in the TEM images corresponds with the known physiology of *S. alvi*, while the colonization dynamics matched our experimental design: Initially, no viable bacterial cells were detected when plating homogenized guts, and we observed no bacterial cells in the TEM images of gut sections (See Dataset S2). Subsequently, we observed an exponential increase in bacterial numbers over the following 72 hours. While the number of cells observed in TEM images did intuitively seem low to us as well, they are within expected ranges based on the CFU plating. Given an approximate ileum length of 2 mm and assuming that the bacteria are homogeneously distributed, then with tissue sections of 100 nm, a bacterial load of $\sim 10^6$ - 10^7 bacteria in the ileum would result in ~ 50 - 500 bacterial cells per tissue section. We counted bacterial cells in a select number of TEM images and observed for example, ~ 360 bacteria in a 48h section and ~ 1100 bacteria in a 72h section which is within the same order of magnitude as our back of the envelope calculations above.

Please indicate in legend EAA and NEAA – although very cool to see the essential amino acids unlabelled and other metabolites labeled at approx. 50%. Although the percentage of organic acids surprises (I expected much high labelling after the glucose pulse).

Reply: The acronyms have been included in the Figure legend. We suspect that the lower labelling fraction after an extended ^{13}C glucose pulse results from dilution by host catabolism of (unlabeled) biomass. Bees do not grow significantly after emerging as adults and thus most of the sugar they ingest is likely catabolized for energy (and released as CO_2) rather than assimilated as new biomass. While it is a fortuitous advantage that the bees can survive on sucrose solution for many days, this is a metabolically deficient diet and they probably compensate by recycling previously synthesized biomass to meet demands for limiting nutrients. This would be reflected in a high reflux of unlabeled carbon into the measured organic acids and amino acids. Such an isotopic dilution has been quantitatively

21evaluated in plant cultures but not to our knowledge in animal models [Nargund et al, Mol. Biosyst.].

In dataset1 the % labeling is very different within the group of NEAA, I would not summarize them all as one group average, rather show exemplary (maybe a high and low example)

Reply: The figure has been updated to show labelling of individual compounds.

The supplement figure S4 indicates high consumption of certain carbon sources, also caffeate. This compound is listed also in fig4B and its not clear if growth happened or not, also not obvious why its indicated as carbon source.

Reply: Caffeate was depleted in the *in vitro* experiment (updated Figure S6) but did not result in an increased growth rate (Figure 4B). Caffeate is one of the pollen associated molecules depleted by *S. alvi* *in vivo* (Figure 1C, Figure S3), which is why it was chosen as a growth substrate *in vitro*.

Discussion is very well balanced and carefully phrased statements including naming of limitations. The discussion around L287: nutrients of pollen have minor impact compared to host. Which data supports this claim? Is there 22anoSIMS data on *S. alvi* after the pulse of ¹³C glucose plus unlabelled pollen?

Reply: We based this claim on our colonization data, which showed little to no increase in colonization when pollen was added to the diet. We have updated the text to make this clear Line 79. We now also have shown with a new experiment that the abundance of key *S. alvi* substrates like citrate and 3Hmg is substantially lower in pollen than in the gut (Figure 2A).

Small points:

L45 - These phylotypes are culturable – all of the 8-10?

Reply: Yes, they are all culturable and can be colonized into gnotobiotic honey bees [Kesnerova et al., PLoS Biol]

L50 – starches – I think starch is one sugar polymer and does not summarize different types

Reply: We have modified “starches” to “starch” in the text

L54 – biomass through gluconeogenesis. provide ref please

Reply: Reference [10] [Kwong et al., PNAS] has been added.

L68 – these compounds – I think you can be already more specific here

Reply: We have replaced “these compounds” with “carboxylic acids”.

L106 – and an unknown – specify the unknown here (unknown_RT)

Reply: Changed. Thank you.

L136 – synthesized in adult bees – should be a statement like: not synthesized during the course of experiment

Reply: The text was updated to: “The lack of isotope labelling indicates that these compounds were not actively synthesized from glucose by adult bees during the first six days post-emergence. Instead, they were either leftover in the gut from the larval development stage or were acquired from the larval diet, such as in the case of essential amino acids and their catabolic products.” Line 154

L140 – I like the careful statement and explanation of possible source of isotope dilution

Reply: Thank you.

Caffeic acid or caffeate, authors should check the Ms. for the nomenclature of all metabolites and try to be consistent. In a few places its off.

Reply: Thank you

The authors did very well on reporting the replication and provide sufficient datasets. The GitHub link to the raw data did not work for me and I would like to see the metabolomics data deposited on metabolights or similar database.

Reply: Thank you for pointing out this issue. The Github link has been fixed. Additionally, metabolomics data have been deposited to the EMBL-EBI MetaboLights database (DOI: 10.1093/nar/gkz1019, PMID:31691833) with the identifier MTBLS8058. Please note, the MetaboLights link may not activate to the public until it has been reviewed and curated by MetaboLights. <https://www.ebi.ac.uk/metabolights/MTBLS8058>

Reviewer #3 (Remarks to the Author):

Review of Quinn et al: Foraging on host synthesized metabolites enables the bacterial symbiont *Snodgrassella alvi* to colonize the honey bee gut

This paper by Quinn et al., seeks to understand whether host-produced metabolites feed *Snodgrassella*, a gut symbiont of honey bees. The authors tackle their question using a series of approaches that include manipulative diet experiments coupled with metabolome analysis, stable isotope labelling experiments, NanoSIMS, in vitro growth assays, examination of genetic variation in the colonization success of *Snodgrassella* in honey bee guts under a range of dietary conditions. Taken together the data show that *Snodgrassella* feeds on host synthesized metabolites, a result that has exciting implications for understanding the establishment and evolution of this symbiosis.

Reply: Thank you for the kind summary of our study.

I have two main concerns about the work.

1/ qPCR quantification of *Snodgrassella* and other symbionts population sizes rests on the assumption that the bacteria are not polyploid. It is common for symbiont population size and symbiont genome copy number to be uncoupled because the ploidy of symbionts is not fixed or constant see:

Komaki, K., and Ishikawa, H. (2000). Genomic copy number of intracellular bacterial symbionts of aphids varies in response to developmental stage and morph of their host. *Insect. Biochem. Mol. Biol.* 30, 253–258. doi: 10.1016/s0965-1748(99)00125-3

Mergaert, P., Uchiumi, T., Alunni, B., Evanno, G., Cheron, A., Catrice, O., et al. (2006). Eukaryotic control on bacterial cell cycle and differentiation in the Rhizobium-legume symbiosis. *Proc. Natl. Acad. Sci. U.S.A.* 103, 5230–5235. doi: 10.1073/pnas.0600912103

Woyke, T., Tighe, D., Mavromatis, K., Clum, A., Copeland, A., Schackwitz, W., et al. (2010). One bacterial cell, one complete genome. *PLoS One* 5:e10314. doi: 10.1371/journal.pone.0010314

The variable ploidy of symbionts means that it is impossible to obtain meaningful estimates of symbiont population sizes using qPCR unless the authors demonstrate that *Snodgrassella* are haploid.

Given that the authors also use culture-based methods of quantifying *Snodgrassella* populations, I suggest dropping the qPCR estimates from the manuscript. The overall results will not be changed by dropping the qPCR data.

Reply: Intracellular bacteria such as *Sulcia muelleri* can indeed be polyploid with 200-900 genome copies per cell [Woyke et al, PLOS ONE]. A few cases of extracellular bacteria with large cell size (100-300 μm) have also been shown to have polyploidy genomes [Mendell et al, PNAS]. In contrast to these extreme examples, *Snodgrassella alvi* is a Betaproteobacterium with an average cell size of 1 μm . Polyploidy has not been reported for any of the related bacteria of *S. alvi*. Moreover, estimates of bacterial load determined with qPCR is typically in the range of estimates based on CFU. Therefore, we can be pretty certain that *S. alvi* is haploid and qPCR provides a robust analysis to estimate bacterial loads. However, it is true that the target we have been using for the qPCR assay (i.e. the 16S rRNA gene) is present in four copies in both the genome of *S. alvi* and all *Gilliamella* strains (See Table S1). We have considered this information in our data analysis pipeline when converting from CT values to genome equivalents by dividing the copy number counts by four to infer number of genome equivalents. It is important to note that one can also argue against using CFU to determine absolute bacterial loads, as single cells tend to aggregate forming colonies that originate from more than 1 cell in many cases and resulting in wrong estimates of the total bacterial load. Additionally, the CFU data cannot resolve the absolute amounts of *S. alvi* and *Gilliamella* in our co-colonization conditions. Therefore, we stand by the validity of our approach and believe that the qPCR results provide complimentary confirmation of our CFU data.

2/Presentation of the metabolome data and analyses need to be improved. To make it more accessible to people unfamiliar with these types of analyses. I found the work presented in Figure 1C particularly hard to understand with respect to the directionality of the relationships/contributions. I am not convinced that these data are needed to address the central question of the paper as it is currently presented.

Reply: We have updated the text and figure legend to increase clarity for the reader. Specifically, we show now arrows which highlight the directionality of metabolite changes. Previous work in the field suggested that metabolic cross-feeding from the *Gilliamella* species was responsible for colonization by *S. alvi*. We designed our experiments to detect this, and therefore we feel it is important to address the effects of diet and *Gilliamella* on *S. alvi* colonization. Our multi-factorial design is best analyzed by mixed linear models which can untangle effects of multiple variables on metabolite abundances.

The most compelling results are the labelling experiments.

Reply: Thank you! We believe so too.

Detailed comments on the manuscript to assist in improving the work:

- Line 73 – it would be good to give a brief description of how microbiota depleted honey bees were generated. I also suggest avoiding use of the abbreviation MD – and just refer to them as microbiota depleted through the manuscript

Reply: We refer you to our response to Reviewer #1 on the issue of generating microbiota free bees. We have changed the notation in the text and figures from Microbiota-depleted (MD) to Microbiota-free (MF). We also now provide more details on how they have been generated. See section beginning at Line 331.

- Line 74 -75 : reword “alone or with both” this is confusing

Reply: We have revised the text. It now reads as follows: “*We colonized microbiota-free honey bees with S. alvi (strain wkB2) or a mixture of Gilliamella strains (strains wkB1, ESL0169, ESL0182, ESL0297) or with both phylotypes together*” See line 74.

- Line 75 and throughout – pollen, from what plant? I assume that this matters and that the metabolic profile of pollen from different plants varies. If not, state that all pollen has the same metabolic profile.

Reply: We utilize a mixture of polyfloral pollen. The methods have been updated to indicate the source of the pollen and how it was sterilized. See line 343.

- Line 135: it would be good in here and likely other places to be clear when something is not expected, for example, the result that essential amino acids are not enriched is entirely unsurprising because bees can't make essential amino acids. I suggest looking carefully through the whole manuscript to be clearer about what the null expectations for each experiment are as this will help the reader understand better when a result is consistent with expectations and when it is not.

Reply: Thank you. We have revised the text accordingly. Examples include line 78, line 115, line 157, and line 189.

- Line 321 – what is the source of the pollen?

Reply: The methods have been updated to indicate the source. See line 343.

- Line 333 – frozen at what temperature?

Reply: The text has been updated to: "The third aliquot was frozen at -80 °C until DNA extraction."

- Line 337 – explain that U-13C is universally labelled, what does that mean?

Reply: We have updated the text to state that U-13C is universally ¹³C labelled glucose. See line 391. This means that all 6 carbon atoms of glucose are ¹³C rather than ¹²C.

- Line 341 – was the ¹²C glucose also universally labelled?

Reply: ^{12}C glucose contains the natural abundance of ^{12}C and ^{13}C , i.e. ~98.9% of all C atoms are ^{12}C . The text has been updated to: "naturally abundant (i.e. ~98.9% ^{12}C) glucose". See Line 395.

- Line 358 – replace "an" with "a"

Reply: Done.

- Line 363 – italics for species name

Reply: Done.

- Line 367 and many other places – use the micro symbol not a letter u

Reply: Done.

- Line 390 and other places – mL not ml

Reply: Done.

- Line 463 – replace , with . in number

Reply: Text updated to "13,300 g".

- Figure 1A – significance missing for 3rd data set across

Reply: The 3rd condition (*S. alvi*, Pollen (-)) is the reference condition that other conditions are compared to.

- Fig 1C – needs improved labelling and explanation to increase the accessibility of these data to the reader. E.g. for cross-feeding include an indication of the direction in the figure. Better label the figure so you don't have to read the legend to find out what the red vs. blue dots mean. The dotted lines are light, maybe shade the area within the lines? Line 517 – lower-left section – maybe also shade this area??

Reply: The Figure has been updated to improve clarity for the reader. Specifically, we have indicated how the direction of metabolite changes corresponds to the effect sizes along the x- and y axes as well as the data colors.

- Fig 3B – line is too faint

Reply: The Figure has been updated to improve clarity for the reader-

- Fig 3E - crop is invisible

Reply: The Figure has been updated to improve clarity for the reader-

- Fig 4B – add a key to the figure on the figure

Reply: Figure 4B has been updated to fix the formatting error.

- Line 562 – did you perform a correction for multiple tests??

Reply: Yes, the p-values were adjusted using the Benjamini-Hochberg correction. We have now stated this in the text.

- Fig 5A – add a bar across the top marking those that are native to Apis and those that are native to Bombus.

Reply: The Figure has been updated to improve clarity for the reader-

- Fig 5B – the colors of 1 and 5 are too similar

Reply: The Figure has been updated to improve clarity for the reader-

- Fig 6 – blue arrows are substrates? Aren't they enzymes? Substrates in boxes??

Reply: Arrows represent metabolite fluxes, which are sometimes enzyme catalyzed. Blue arrows show the entry points of host synthesized metabolites into the metabolic network of *S. alvi*-

- Line 588 – in what genome?

Reply: This scheme and corresponding caption depicts *S. alvi*'s metabolism based on the genome of the type strain wkB2.

- Fig S2 – text in panels is too small to read

Reply: Text in the Figure has been updated to conform to journal guidelines.

- Fig S4 – for glucose at 0% the dots are aligned on the horizontal to indicate that three data points were taken. Is this also true for those substrates that reach 100% or do these represent just a single data point?

Reply: "Bars represent the average value from 3 biological replicates (black dots)."

- Figure S5 – names of taxa too small to read

Reply: Text in the Figure has been updated to conform to journal guidelines

Decision Letter, first revision:

Message: Our ref: NMICROBIOL-23010206A

12th October 2023

Dear Philipp,

Thank you for submitting your revised manuscript "Foraging on host synthesized metabolites enables the bacterial symbiont *Snodgrassella alvi* to colonize the honey bee gut" (NMICROBIOL-23010206A). It has now been seen by the original referees and their comments are below. The reviewers find that the paper has improved in revision, and therefore we'll be happy in principle to publish it in Nature Microbiology, pending minor revisions to satisfy the referees' final requests and to comply with our editorial and

30formatting guidelines.

Thank you again for your interest in Nature Microbiology Please do not hesitate to contact me if you have any questions.

Sincerely,

[redacted]

Reviewer #1 (Remarks to the Author):

The authors have responded to my concerns by clarifying their text, adding detail where needed (for example, the medium recipe) and additional experimentation. I am satisfied with their response to reviewer critiques and congratulate them on their manuscript.

Reviewer #2 (Remarks to the Author):

Many thanks for the explanations and answers to all reviewer comments. I like that Quinn et al. are careful with the data interpretation and explain why they only report e.g. isotopic enrichment and do not compare metabolite abundances between experiments. One of my criticism regarding the link between nanoSIMS signal and microscopy is solved after the explanations, one can follow the arguments and it confirms the claims made. The updated figures are now better to understand with all the data presented!
All other points answered and changed - no further comments or suggestions from my side.

Edits:

5 μm (not μM) scale bar in legend of figure 3

Final Decision Letter:

31Mess 30th November 2023

age:

Dear Philipp,

I am pleased to accept your Article "Host-derived organic acids enable gut colonization of the honey bee symbiont *Snodgrassella alvi*" for publication in Nature Microbiology. Thank you for having chosen to submit your work to us and many congratulations.

Acceptance of your manuscript is conditional on all authors' agreement with our publication policies (see <https://www.nature.com/nmicrobiol/editorial-policies>). In particular your manuscript must not be published elsewhere and there must be no announcement of the work to any media outlet until the publication date (the day on which it is uploaded onto our website).

Please note that *Nature Microbiology* is a Transformative Journal (TJ). Authors may publish their research with us through the traditional subscription access route or make their paper immediately open access through payment of an article-processing charge (APC). Authors will not be required to make a final decision about access to their article until it has been accepted. [Find out more about Transformative Journals](https://www.springernature.com/gp/open-research/transformative-journals)

Authors may need to take specific actions to achieve [compliance](https://www.springernature.com/gp/open-research/funding/policy-compliance-faqs) with funder and institutional open access mandates. If your research is supported by a funder that requires immediate open access (e.g. according to [Plan S principles](https://www.springernature.com/gp/open-research/plan-s-compliance)) then you should select the gold OA route, and we will direct you to the compliant route where possible. For authors selecting the subscription

32publication route, the journal's standard licensing terms will need to be accepted, including [self-archiving policies](https://www.nature.com/nature-portfolio/editorial-policies/self-archiving-and-license-to-publish). Those licensing terms will supersede any other terms that the author or any third party may assert apply to any version of the manuscript.

With kind regards,

P.S. Click on the following link if you would like to recommend Nature Microbiology to your librarian <http://www.nature.com/subscriptions/recommend.html#forms>

** Visit the Springer Nature Editorial and Publishing website at http://editorial-jobs.springernature.com?utm_source=ejp_NMicro_email&utm_medium=ejp_NMicro_email&utm_campaign=ejp_NMicro for more information about our career opportunities. If you have any questions please click [here](mailto:editorial.publishing.jobs@springernature.com).**